# Amyloid precursor protein modulates cerebellar Purkinje cell activity and motor function through regulation of Nav1.6 currents

Miao-Jin Ji[1,☯], Tong-Xuan Wu[1,☯], Chenhao Tian[1], Xiang Cao[1], Ruyuan Wei[1], Yin-Yin Yang[1], Xinran Meng[1], Huanyao Tang[1], Tiantao Cui[1], Jiao Yang[1,2], Xin Tang[1], Chao Liu[1]*

1 Jiangsu Province Key Laboratory of Anesthesiology, Jiangsu Province Key Laboratory of Anesthesiology and Brain Science, NMPA Key Laboratory for Research and Evaluation of Narcotic and Psychotropic Drugs, School of Anesthesiology, Xuzhou Medical University, Xuzhou, China, 2 Department of Anesthesiology, Zhongshan Hospital, Fudan University, Shanghai, China

☯ These authors contributed equally to this work.

* Chaoliu@xzhmu.edu.cn

## Abstract

Amyloid precursor protein (APP)-null mice exhibit significant deficits in motor performance, including reduced grip strength and impaired locomotion; however, the underlying neurophysiological mechanisms remain unclear. In this study, we show that conditional knockdown of APP selectively in Purkinje cells (PCs) recapitulates these motor deficits, while exogenous expression of APP in APP-null mice rescues motor function. Electrophysiological analysis revealed that APP deficiency leads to aberrant firing patterns in PCs and reduces inhibitory synaptic transmission onto neurons of the deep cerebellar nucleus (DCN). We identified a marked reduction in Nav1.6-mediated sodium currents as the key mechanism underlying abnormal action potential firing and propagation in APP-deficient PCs. Importantly, all electrophysiological and behavioral deficits were rescued by PC-specific APP reconstitution. These findings reveal a novel and essential role for APP in cerebellar motor control by regulating Nav1.6 channel activity and PC excitability.

## Introduction

Amyloid precursor protein (APP) is a type I transmembrane protein processed by secretases to generate β-amyloid peptides and other fragments. While APP is best known for its central role in Alzheimer's disease pathogenesis, its physiological functions remain incompletely understood. Recent studies have demonstrated that APP plays critical roles in neuronal development [1,2], synaptic function [3], and intracellular signaling [4]. These processes influence a wide range of central nervous system-regulated functions, including learning, memory [5–7], and motor functions [8–10]. The APP-null mice exhibited impaired motor performance, learning capacity, and poor memory [5–7]. While the role of APP in cognitive

**Data availability statement:** All relevant data are within the paper and its Supporting information files.

**Funding:** This work was supported by the National Natural Science Foundation of China (https://www.nsfc.gov.cn/) (32271010 to CL; 81974157 to CL), the Fund for Jiangsu Province Specially-Appointed Professor (to CL), the Starting Grant for Excellent Talents in Xuzhou Medical University (https://www.xzhmu.edu.cn/) (to CL), and the Innovative Youth Team Project in Xuzhou Medical University (https://www.xzhmu.edu.cn/) (TD202203 to MJJ). The funders had no role in study design, data collection and analysis, decision to publish, or preparation of the manuscript.

**Competing interests:** The authors have declared that no competing interests exist.

**Abbreviations:** AHP, afterhyperpolarization potential; AICD, APP intracellular domain; AP, action potential; APP, amyloid precursor protein; co-IP, co-immunoprecipitation; DCN, deep cerebellar nucleus; FBS, fetal bovine serum; LCM, laser capture microdissection; mIPSC, miniature inhibitory post-synaptic currents; OCT, optimal cutting temperature compound; PBS, phosphate-buffered saline; PCs, Purkinje cells; PFA, paraformaldehyde; qPCR, quantitative PCR; rAAV, recombinant adeno-associated virus; SSs, simple spikes; sIPSCs, spontaneous inhibitory post-synaptic currents; WT, wild-type.

impairment has been extensively studied, its involvement in motor control remains less explored. Interestingly, in contrast to other movement disorder models, APP-null mice specifically display deficits in locomotor activity and grip strength [8,9]. Initial hypotheses suggested that APP loss disrupts neuromuscular junctions. However, APP single-knockout mice exhibited no significant alterations in neuromuscular transmission or synaptic morphology at the neuromuscular junctions [11], suggesting that APP may instead regulate higher-order motor control circuits rather than the function of neuromuscular junctions. In mammals, voluntary movement is orchestrated by the corticospinal tract, a motor pathway connecting the motor cortex to lower motor neurons and involving multiple regions, including the motor cortex, subcortical nuclei, cerebellum, brainstem, and spinal cord [12]. Conditional APP knockout under the Nex promoter—driving Cre expression in corticospinal tract-associated regions, including the motor cortex, superior and inferior colliculi, medulla oblongata, pons, cerebellar granule cell layer, and deep cerebellar nuclei (DCNs) [13], does not impair grip strength or locomotor activity [7], suggesting that APP in these brain regions is unlikely to account for its role in motor function regulation.

The cerebellum is a crucial subcortical region involved in motor control, comprising the cerebellar cortex, DCNs, and medulla. Purkinje cells (PCs) in the cerebellar cortex receive input from mossy fibers and climbing fibers, integrating these signals and projecting to DCN neurons. As the sole output neurons of the cerebellar cortex, PCs are critical for encoding motor function. In humans, cerebellar activity increases linearly with increasing grip force in the ipsilateral limb [14]. Patients with cerebellar lesions display muscle force production deficits [15]. The action potential (AP) firing of PCs is spontaneous and of high frequency, dependent on voltage-gated sodium channels. In the cerebellar cortex, Nav1.6 and Nav1.1 are predominant in PCs, while Nav1.2 is primarily expressed in cerebellar granule cells [16], as demonstrated by morphological [17] and electrophysiological studies [18]. Nav1.6 accounts for ~70% of persistent sodium currents and ~82%–92% of resurgent sodium currents in PCs [18]. Three types of sodium currents are typically studied in PCs: resurgent currents, responsible for the generation of spontaneous firing; transient sodium currents, contributing to the upstroke of APs, and persistent sodium currents, which regulate plateau potentials and amplify dendritic input [19].

APP is abundantly expressed in cerebellar PCs [20]. Our prior study showed that APP increases Nav1.6 currents by enhancing its cell surface localization in vitro [21]. However, whether APP regulates the physiological function of cerebellar PCs and participates in motor control remains unclear. In this study, we investigated the role of APP in cerebellar PCs by analyzing the motor function deficits, PC firing patterns, and cerebellar circuit dysfunctions in APP-null mice. We found that APP in cerebellar PCs is critical for PC firing through the regulation of Nav1.6 and plays an essential role in PC-mediated motor control.

## Results

### Cerebellar PC APP mediates motor functions in mice

Consistent with previous study [8,9], we observed that APP-null (*App*$^{-/-}$) mice exhibited reduced locomotor activity (S1A Fig) and weaker forelimb grip strength (S1B Fig) compared with wild-type (WT, *App*$^{+/+}$) littermates. Further, animals were tested for their capacity for motor learning and coordination using a rotarod and footprint test. APP-null mice showed normal motor learning over 3 days of training and their latency to fall was similar to that of WT mice in the test day (S1C Fig). During the footprint test, there was no differences between APP-null and WT mice in sway length and the overlap, while the stride length and the stance length were all significantly lower in APP-null group compared with that of WT (*App*$^{+/+}$) group (S1D and S1E Fig). The reduced stride length in footprint test (S1E Fig) likely reflects smaller body size in APP-null mice (S1F Fig). To control for this confounder, we performed the balance beam assay. No significant differences were observed in traversal time or number of errors on 12- or 6-mm beams (S1G Fig), indicating intact gross motor coordination in APP-null mice. Thus, APP deletion specifically impairs open field locomotion and forelimb grip strength, independent of anthropometric differences.

APP is abundantly expressed in PCs of adult WT mice (Fig 1A). Given that cerebellar PCs play a critical role in motor function, we hypothesized that the deletion of PC-specific APP could be the primary cause of the motor dysfunction observed in APP-null mice. To specifically evaluate the effect of PC APP on motor performance, we injected a lentivirus encoding human APP695 with mCherry (LV-*CAMK2A*-SP-Flag-APP695-IRES2-mCherry) into the cerebellar cortex (Fig 1B). The *CAMK2A* promoter, which has been shown to drive gene expression specifically in PCs [22], was used in this experiment. Expression of mCherry was observed exclusively in the Calbindin-positive PCs (Fig 1B). Laser capture microdissection (LCM) coupled with Reverse Transcription Quantitative PCR (RT-qPCR) analysis further confirmed a significant increase of the APP expression in PCs (Fig 1C). PC-specific expression of APP in APP-null mice significantly improved their performance in locomotor activity (Fig 1D) and forelimb grip strength test (Fig 1E), but didn't affect their body size (Fig 1F).

To further investigate the contribution of APP to motor control, we utilized mice with conditional APP knockdown restricted to PCs. We achieved PC-specific *App* knockdown by co-injecting WT mouse cerebellar cortex with AAV2/9-L7-Cre [23] and AAV2/9-DIO-(U6-EGFP)-sh*App* [24], whereby Cre recombinase activates the expression of shRNA targeting *App* specifically in L7-positive cells. Immunofluorescence confirmed efficient PC-specific transduction mediated by AAV under the L7 promoter, as shown by its colocalization with calbindin (Figs 1G and S2). LCM coupled with RT-qPCR verified dramatic APP downregulation in transduced neurons (Fig 1H). Subsequent behavioral analyses revealed significant motor deficits in conditional knockdown mice, including reduced ambulation in open field tests (Fig 1I) and impaired forelimb grip strength (Fig 1J).

Taken together, these results showed that PC-specific APP expression mediates cerebellar-dependent motor performance.

### Full-length APP is necessary for rescue of motor function deficits in APP-null mice

Proteolytic processing of full-length APP generates bioactive fragments including sAPPα (sufficient for spatial learning and long-term potentiation rescue in APP-null mice) [9] and APP intracellular domain (AICD) (with transcriptional activity). To determine whether the intact holoprotein is required for motor functions, we delivered holo-APP, sAPPα, or AICD to APP-null mice via cerebellar lentivirus injection. Fluorescence imaging confirmed targeted expression in PCs (Fig 2A). LCM-coupled RT-qPCR verified successful APP reconstitution in transduced neurons (Fig 2B). Behavioral analyses demonstrated that only full-length APP (holo-APP) significantly rescued motor deficits, exhibiting significant improvement in open field ambulation (Fig 2C) and grip strength (Fig 2D) in APP-null mice. sAPPα and AICD provided no significant rescue (Fig 2C and 2D). These results establish that full-length APP is necessary for cerebellum-mediated motor function.

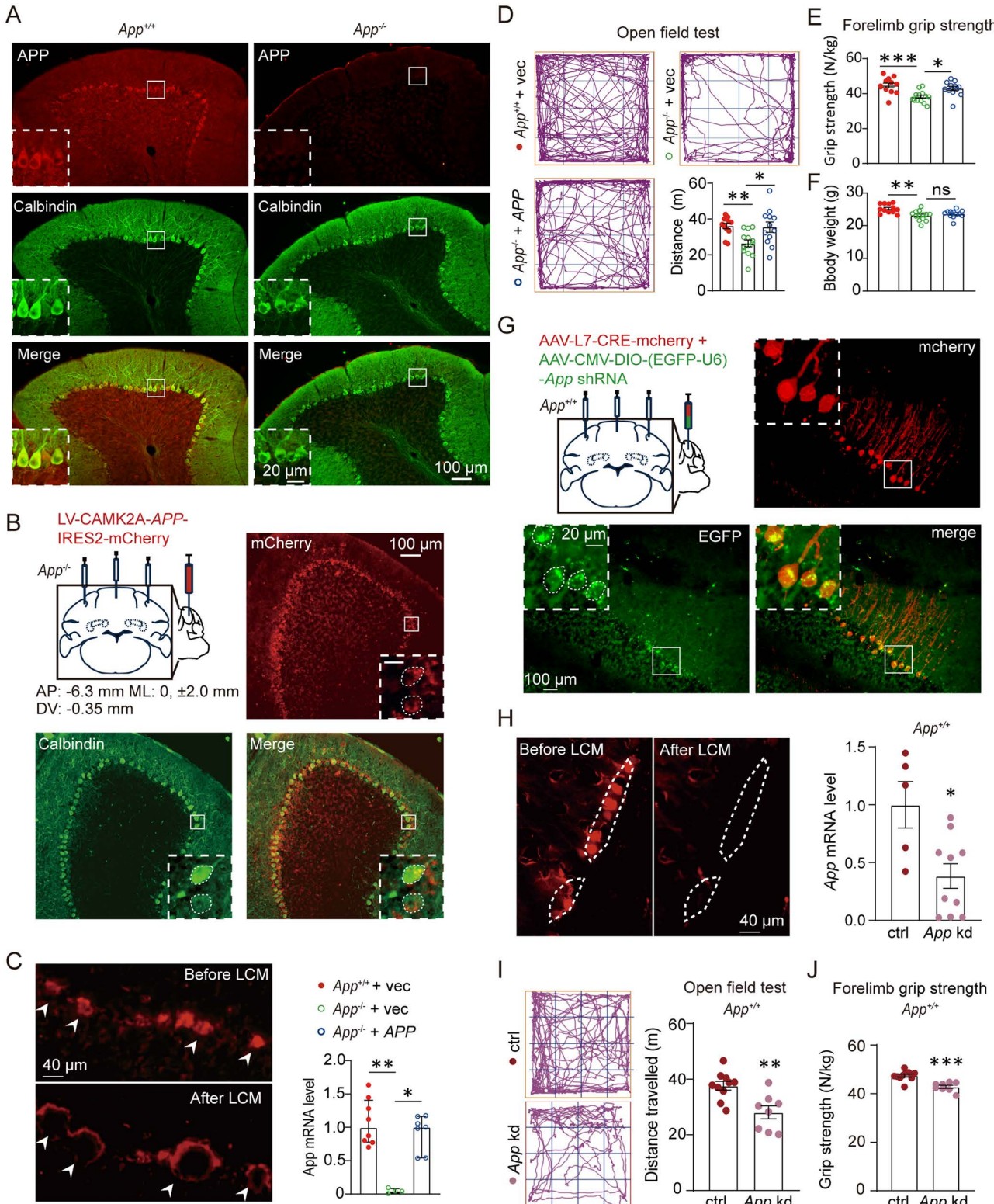

**Fig 1. Cerebellar Purkinje cell (PC)-specific amyloid precursor protein (APP) expression modulates motor function. (A)** APP (red) and calbindin (green) immunofluorescence in 20-μm-thick cerebellar sagittal sections. The APP immunofluorescence in wild-type (*App*⁺/⁺) or APP-null (*App*⁻/⁻) mice

were acquired under identical imaging settings. Insets: magnified views. Scale bars: 100 μm (main), 20 μm (insets). The PC contours are indicated by dashed lines. **(B)** Lentivirus-mediated overexpression of APP695 in the cerebellar cortex of $App^{-/-}$ mice. Representative images showing co-expression of APP/mCherry (red) and the PC marker calbindin (green) in a sagittal section of cerebellum. Scale bars: 100 μm (main); 20 μm (insets). **(C)** Left: representative fluorescence images showing before or after laser capture microdissection (LCM) of LV-transduced PCs (mcherry⁺) from fresh-frozen coronal sections. Right: $App$ mRNA levels in dissected cells detected by qPCR (Data presentation: median with interquartile range. $App^{+/+}$+vec, $n=8$; $App^{-/-}$+vec, $n=4$; $App^{-/-}$+APP, $n=7$ samples; Kruskal–Wallis test). Scale bar: 40 μm. **(D–F)** Motor phenotyping of $App^{-/-}$ mice with Purkinje APP rescue: (D) Open field total distance (10-min test). (E) Forelimb grip strength. (F) Body weight of all groups of mice. $App^{+/+}$+vec, $n=12$; $App^{-/-}$+vec, $n=13$; $App^{-/-}$+APP, $n=12$. **(G)** Conditional knockdown of $App$ in $App^{+/+}$ mice using a mixed AAV injection strategy. Representative images show the CRE (mCherry⁺) and APP-shRNA (EGFP⁺) expression in PCs in a coronal section of the cerebellum. Scale bar: 100 μm (main), 20 μm (insets). The PC contours are indicated by dashed lines. **(H)** Left: representative fluorescence images before or after LCM of AAV-transduced PCs (mcherry⁺) from fresh-frozen coronal sections of $App^{+/+}$ mice. Right: $App$ mRNA levels in dissected cells detected by qPCR (control, $n=5$, $App$ kd, $n=10$ samples). **(I, J)** Motor deficits following Purkinje APP knockdown: (I) Open field mobility. (J) Forelimb grip strength decline. (Control, $n=10$; $App$ kd, $n=8$ mice). Data presentation: mean with SEM. Statistics: (D–F) one-way ANOVA; (H–J) Unpaired $t$ test; * $P<0.05$; ** $P<0.01$; *** $P<0.001$. The data underlying this Figure can be found in S1 Data.

## APP deficiency impairs PC electrophysiological activity

APP plays a crucial role in neurodevelopment [1,2], so we examined whether APP deficiency impaired the development of PCs. Golgi staining and Sholl analysis revealed that the dendritic branches and spine density were similar between 2-month-old APP-null and WT littermates (S3A–S3C Fig). Further, we compared the morphology of PC in 12-month-old mice. No significant difference in the dendritic branches and spine density was observed between APP-null and WT mice (S3D–S3F Fig). These results indicated that APP deficiency affects neither PC development nor provokes their degeneration.

Next, we investigated whether loss of APP altered the firing patterns of PCs. As established, PCs display three characteristic firing states: silent, simple spikes (SSs), and complex spikes (CSs) [18,25,26]. Most of 22 PCs (19/22, 86.4%) recorded from WT mice exhibited spontaneous firing, while 13.6% (3/22) of PCs maintained silent within 1 h of recording. Complex spikes recorded in 27.3% (6/22) of PCs from WT mice (Fig 3A). The percentage of silent PCs increased to 42.1% (8/19, 42.11%), and no CSs was recorded in 19 PCs from APP-null mice (Fig 3A). We further analyzed the spontaneous firing rate and various AP characteristics, including threshold, peak amplitude, half-width, and afterhyperpolarization potential (AHP) of PCs with SSs in brain slices recorded from APP-null and WT mice. The spontaneous firing rate of PCs in APP-null mice was significantly lower than that in WT mice (Fig 3B). The rheobase of AP in APP-null PC was higher than that in WT mice (Fig 3C). In addition, lower peak amplitude, higher threshold, longer half-width, and bigger AHP (Fig 3D and 3E) were observed in the spontaneous AP spikes of APP-null than those of WT PCs. These results indicate that APP-deficiency significantly impairs the firing properties of cerebellar PCs.

## Reduced Nav1.6, but not Nav1.1-mediated sodium current was detected in APP-null PCs

Consistent with APP enhancing Nav1.6 currents in heterologous systems [21], whole-cell recordings in cerebellar slices revealed altered sodium currents in APP-null PCs. Transient ($I_{NaT}$), persistent ($I_{NaP}$), and resurgent ($I_{NaR}$) currents were evoked using established protocols (S4 Fig) [18]. Given that Nav1.6 and Nav1.1 are the predominant voltage-gated sodium channels in mouse PCs [16], we next sought to determine which of these isoforms was affected by APP deficiency. To this end, we applied subtype-specific blockers/inhibitors: 4,9-anhydrotetrodotoxin (4,9-ah-TTX; selective Nav1.6 blocker) and ICA-121431 (Nav1.1 inhibitor) [19].

Voltage steps (−70 to +10 mV) from a holding potential of −90 mV elicited $I_{NaT}$, mediated by Nav1.6 and Nav1.1 (TTX /4,9-ahTTX + ICA-sensitive, S4A and S4D Fig). APP-null PCs showed reduced peak $I_{NaT}$ but unchanged voltage-dependence (Fig 4A and 4B). Pharmacological dissection revealed selective reduction of Nav1.6-mediated $I_{NaT}$ (Fig 4C), while Nav1.1 currents remained unaffected (Fig 4D).

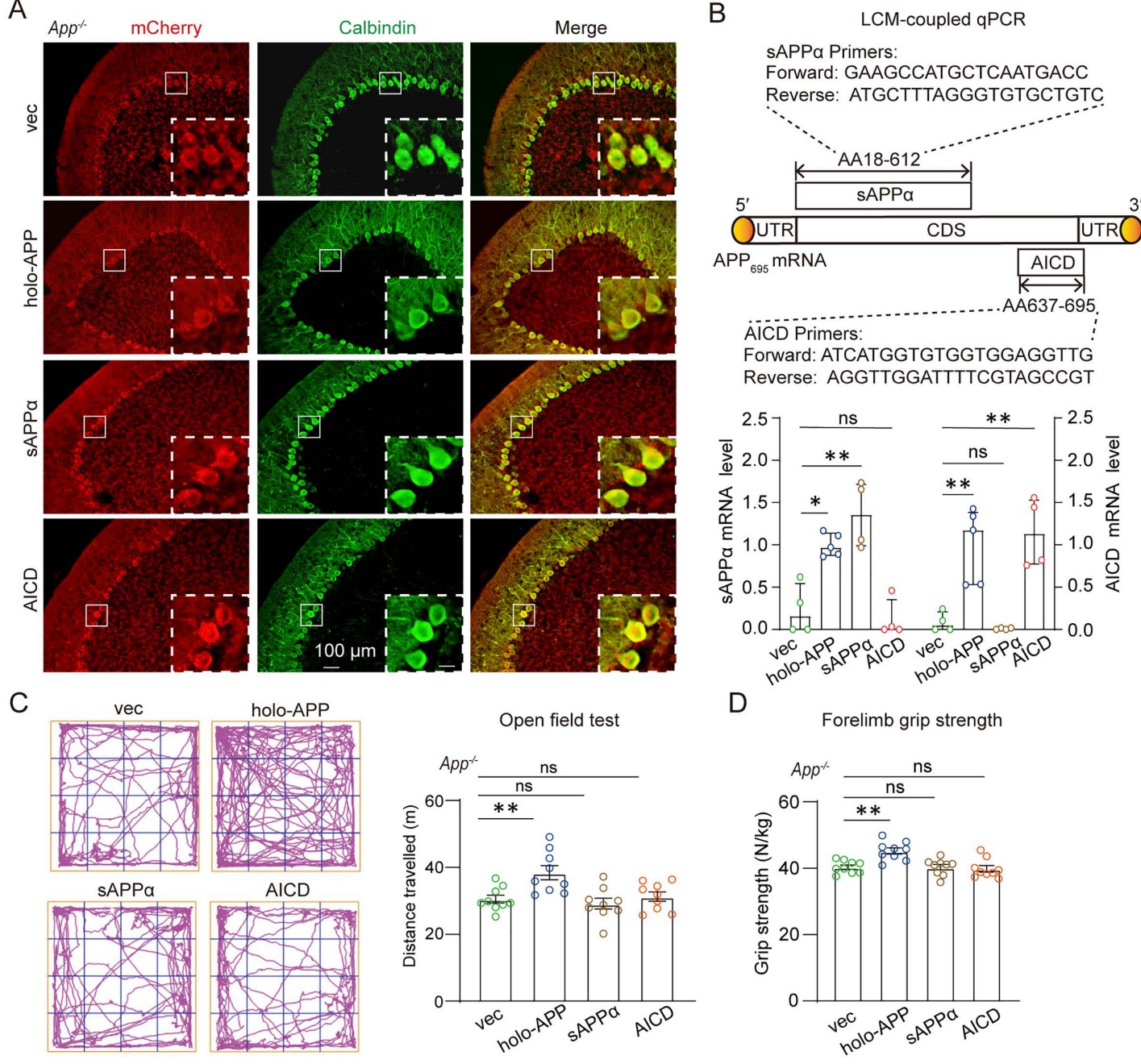

**Fig 2. Full-length APP is required for the rescue of motor deficit in APP-null mice. (A)** Lentivirus-mediated overexpression of full-length APP (holo-APP), sAPPα, or AICD in the cerebellar cortex of *App⁻/⁻* mice. Representative images showing co-expression of APP fragments/mCherry (red) and the PC marker calbindin (green) from coronal sections of the cerebellum. Scale bars: 100 μm (main); 40 μm (insets). **(B)** QPCR analysis of LCMed PCs (mcherry⁺) from fresh-frozen sections. Top: Primer specificity schematic for APP fragments. Bottom: APP fragments mRNA level normalized to *Gapdh*. Data presentation: median with interquartile range. *App⁻/⁻* + vec, $n = 4$; *App⁻/⁻* + holo-APP, $n = 5$; *App⁻/⁻* + sAPPα, $n = 4$, *App⁻/⁻* + AICD, $n = 4$ samples. Kruskal–Wallis test. **(C, D)** Motor phenotyping of *App⁻/⁻* mice with Purkinje APP fragments rescue (n = 9 mice/group): (C) Open field total distance (10-min test). (D) Forelimb grip strength. $N = 9$ for each group. Data presentation: mean with SEM. Statistics: one-way ANOVA. ** $P < 0.01$; ns, non-significant. The data underlying this Figure can be found in S1 Data.

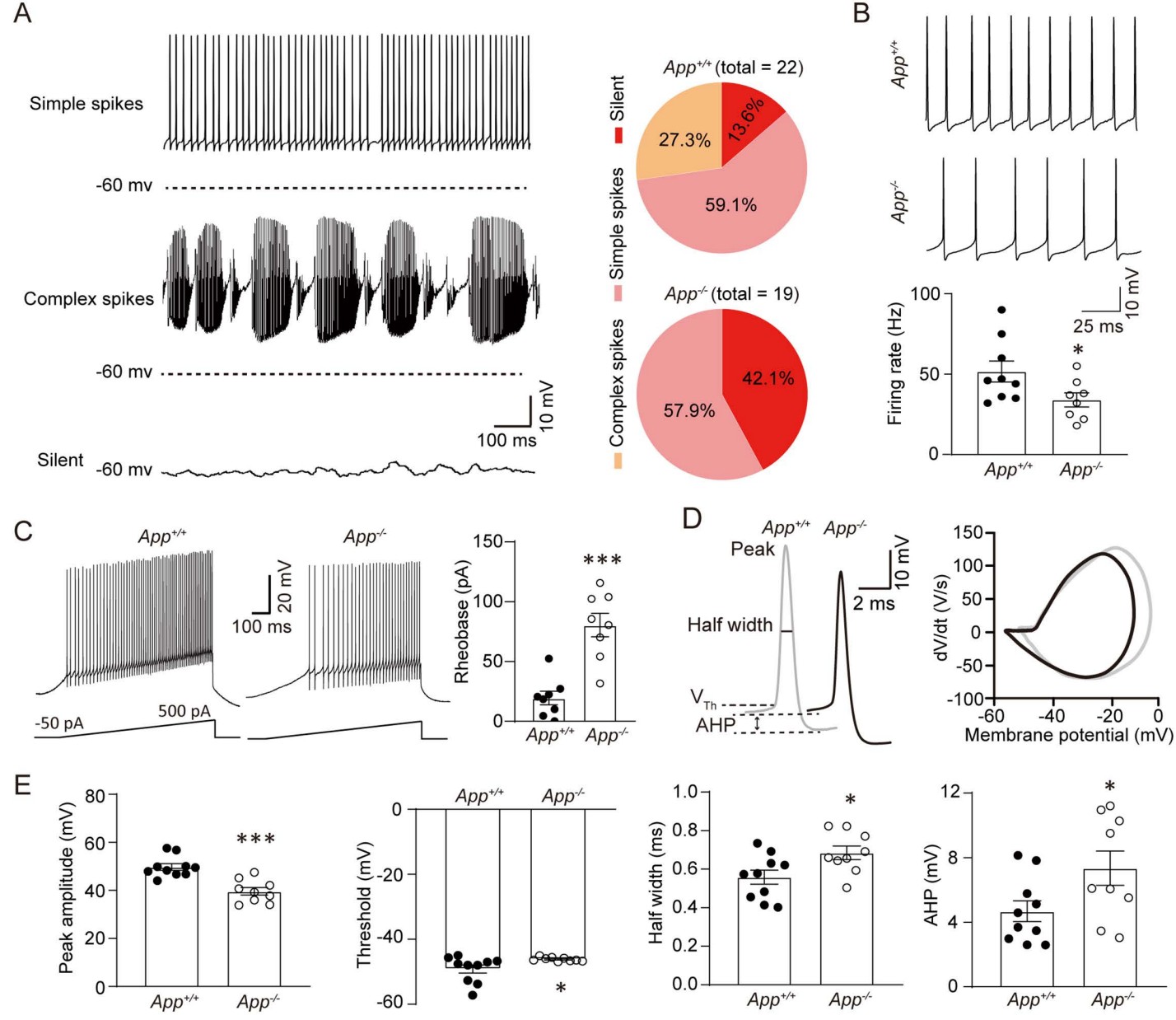

**Fig 3. APP deficiency impairs the normal firing patterns of PCs. (A)** Representative spontaneous firing patterns recorded in PCs of wild-type (*App*$^{+/+}$) mice. Pie charts in the right panel showing the percentage of PC firing pattern. *App*$^{+/+}$, $n=22$; *App*$^{-/-}$, $n=19$ cells. **(B)** Representative spontaneous simple spikes recorded in PCs. Quantification of firing rates analyzed from 1-min recordings during stable whole-cell access (≥10 min after breakthrough). *App*$^{+/+}$, $n=9$; *App*$^{-/-}$, $n=8$. **(C)** Left: Representative voltage responses to depolarizing ramp current injections (−50 to +500 pA over 1 s) recorded PCs under whole-cell current-clamp. Right: Rheobase quantification (minimal current eliciting first action potential). $n=8$ cells/group. **(D)** Representative spontaneous action potentials and the corresponding *dV/dt* curve. **(E)** Statistics of peak amplitude, threshold, half-width, and afterhyperpolarization potential (AHP) of action potentials shown in (D), *App*$^{+/+}$, $n=10$; *App*$^{-/-}$, $n=9$. Scale bars are indicated in the images. Data are represented as the mean with SEM. Student *t* test: * $P<0.05$; *** $P<0.001$. The data underlying this Figure can be found in S1 Data.

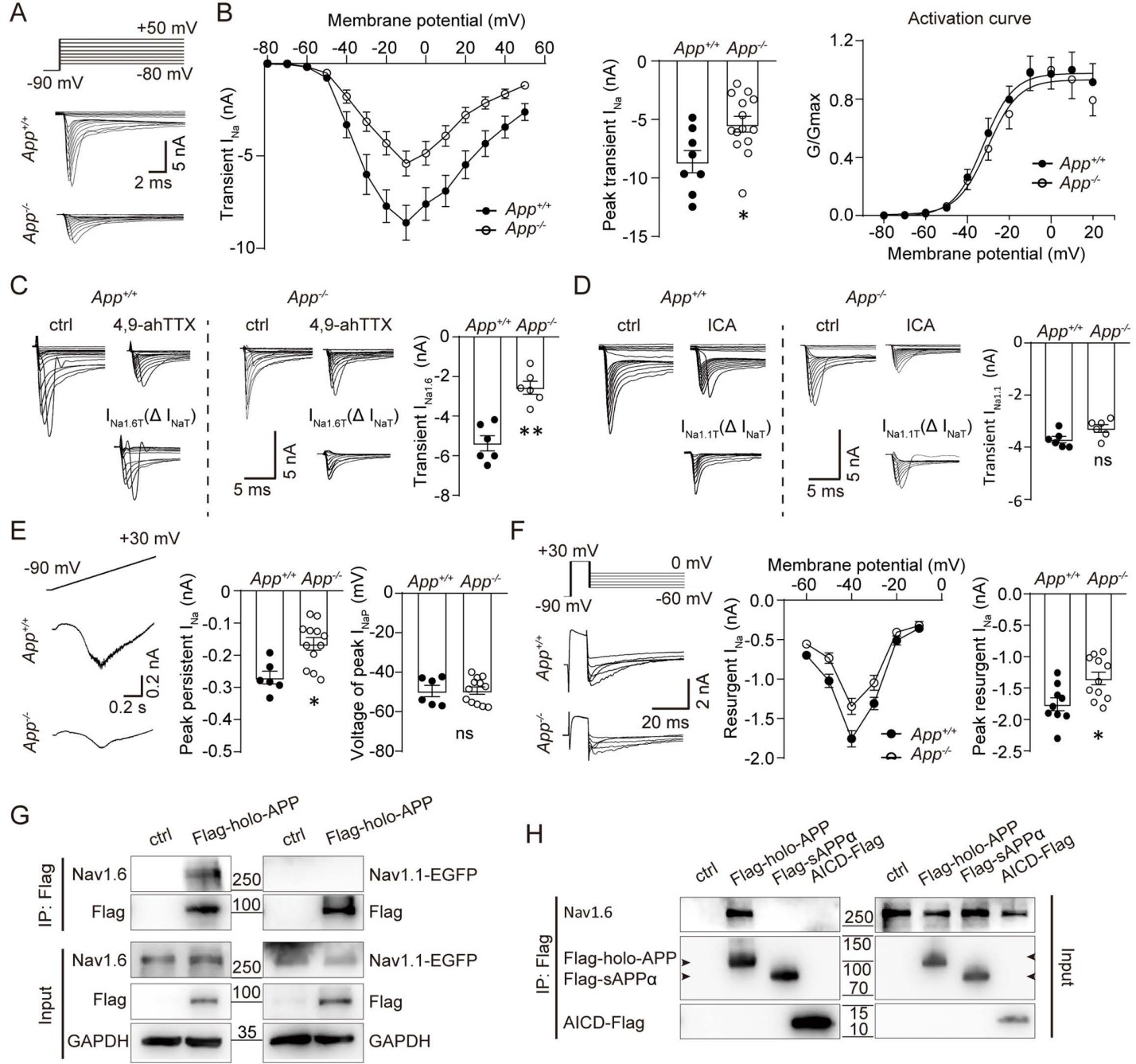

**Fig 4. APP deficiency selectively reduces Nav1.6-mediated Na⁺ currents in cerebellar PCs. (A)** Representative traces for whole-cell recording of transient Na⁺ currents ($I_{Na}$) in PCs of $App^{+/+}$ or $App^{-/-}$ mice. **(B)** Whole cell I–V curve, peak currents, and activation curves of transient $I_{Na}$ as shown in (A). $App^{+/+}$, $n=8$; $App^{-/-}$, $n=14$ cells. **(C, D)** Representative traces and statistical bar-charts showing amplitude of transient $I_{Na}$ ($I_{NaT}$) before or during bath application of (C) Nav1.6-specific blocker 4,9-anhydrotetrodotoxin (4,9-ahTTX, 200 nM) or (D) Nav1.1-specific inhibitor ICA-121431 (ICA, 350 nM). $N=6$ cells for each group. The $I_{Na1.6}$ or $I_{Na1.1}$ were determined by subtracting the sodium currents recorded during bath application of specific blockers/inhibitors from that before drug application. **(E)** Representative persistent Na⁺ currents recorded in PCs. The right panels show statistics of the peak persistent currents and voltage potentials eliciting peak persistent Na⁺ currents. $App^{+/+}$, $n=6$; $App^{-/-}$, $n=12$ cells. **(F)** Representative traces (the left panel) and statistics for whole-cell recording of resurgent $I_{Na}$ in PCs. The middle panel: Whole cell I–V curve of resurgent $I_{Na}$; the right panel: peak resurgent currents. $App^{+/+}$, $n=9$; $App^{-/-}$, $n=11$ cells. Student t test; * $P<0.05$; ** $P<0.01$; ns, nonsignificant. Scale bars are indicated in the images. **(G)** APP co-immunoprecipitated with Nav1.6 but not Nav1.1. HEK293 cells co-expressing Flag-tagged APP and either Nav1.6 or Nav1.1-EGFP were lysed and immunoprecipitated (IP)

with anti-Flag affinity beads (Smart-Lifesciences, #SA042001). Immunoblots (IB) were probed for: Nav1.6 (Alomone Labs #ASC-009), EGFP (Roche Applied Science, #11814460001) or Flag (AlpVHHs, #016-303-005). **(H)** Nav1.6 interacts with full-length APP (holo-APP), but not soluble APPα (sAPPα) and APP intracellular domain (AICD). HEK293 cells co-expressing Nav1.6 and Flag-tagged holo-APP, sAPPα, or AICD were lysed and IPed with anti-Flag affinity beads. IB were probed for Nav1.6 or Flag. Mock transfected controls are shown in the first lane. Input controls (7.5% lysate) shown below/beside corresponding lanes. Data representative of 3 biological replicates. The data underlying this Figure can be found in S1 Data.

Slow ramps (−90 to +30 mV, 0.12 mV/ms) evoked $I_{NaP}$, mediated by Nav1.6 in PCs (blocked by TTX/4,9-ahTTX; S4B, S4E, and S5A Figs). APP-null PCs exhibited diminished $I_{NaP}$ amplitude without activating voltage shift (Fig 4E). Similarly, $I_{NaR}$ (mainly mediated by Nav1.6, evoked by +30 mV and followed by a repolarization to voltage steps between −60 and −10 mV, S4C, S4F, and S5B Figs) was reduced in APP-null PCs with unaltered voltage dependence (Fig 4F).

However, Nav1.6 mRNA (*Scn8a*) levels in PCs did not differ between WT and APP-null mice (S6 Fig), suggesting that APP may not influence *Scn8a* gene expression. Co-immunoprecipitation (co-IP) assays in transfected HEK293 cells (S7 Fig) confirmed a specific physical interaction between APP and Nav1.6 (Fig 4G). This binding was dependent on holo-APP, as neither the soluble ectodomain sAPPα nor the AICD co-precipitated with Nav1.6 (Fig 4H), consistent with the requirement of holo-APP for functional rescue observed in vivo (Fig 2).

Taken together, APP deficiency selectively impairs Nav1.6 but not Nav1.1-mediated sodium currents in PCs through molecular interactions.

## Nav1.6 mediates PC firing and motor deficits in APP-null mice

Our findings align with prior evidence that persistent/resurgent $I_{Na}$ primarily depends on Nav1.6, while transient $I_{Na}$ involves Nav1.1-Nav1.6 synergy [18,27]. Nav1.6 is critical for PC repetitive firing [18]. Accordingly, 4,9-ahTTX abolished repetitive firing in WT PCs (Fig 5A). Conversely, the Nav1.6 positive allosteric modulator poneratoxin (PoTX, inhibits Nav1.6 inactivation, S8 Fig) [28], restored firing in APP-null PCs (Fig 5B). ICA-121431 showed no effect in WT (Fig 5C) or APP-null PCs (Fig 5D).

Intracerebellar infusion studies further implicated the essential role of Nav1.6 in APP-mediated motor function. Histology confirmed cannula placement in cerebellar cortex (Fig 5E). Nav1.6 blocker infusion impaired WT mouse performance in open field and grip-strength tests (Fig 5F and 5G), whereas PoTX rescued motor deficits in APP-null mice (Fig 5F and 5G).

These data provide evidence that APP influences PC firing and Motor function through Nav1.6.

## Exogenous APP expression restored Nav1.6 currents and firing properties of PCs in APP-null mice

We next investigated whether the restoration of the behavioral phenotypes in APP-null mice via exogenous APP expression was linked to the improvement in Nav1.6 function in PCs. Whole-cell patch clamp recordings revealed that transient, persistent, and resurgent $I_{Na}$ were all rescued by exogenous APP expression in APP-null PCs. The peak transient $I_{Na}$ was restored to levels comparable to WT mice (Fig 6A and 6B). APP expression in PCs did not affect the activation curve (Fig 6B). Decreased persistent and resurgent current were also restored by APP expression in PCs (Fig 6C and 6D). These results indicated that exogenous expression of APP in PCs restore the Nav1.6 currents.

In the analysis of firing properties, we found that exogenous APP expression in APP-null PCs restored spontaneous firing patterns nearly to the WT level. Complex spikes were observed in the APP-expressing group (2/21, 9.5%), but not in the control group (Fig 6E). The spontaneous firing frequency of the APP-expressing group was almost restored to that of WT mice (Fig 6F). Additionally, exogenous APP-expression in APP-null PCs restored the rheobase (Fig 6G). Moreover, the spontaneous AP spikes from exogenous APP-expressing PCs displayed higher peak amplitude, lower threshold, shorter half-width, and smaller AHP (Fig 6H and 6I). These data indicate that exogenous APP expression in PCs restored Nav1.6 currents and firing properties in APP-null mice.

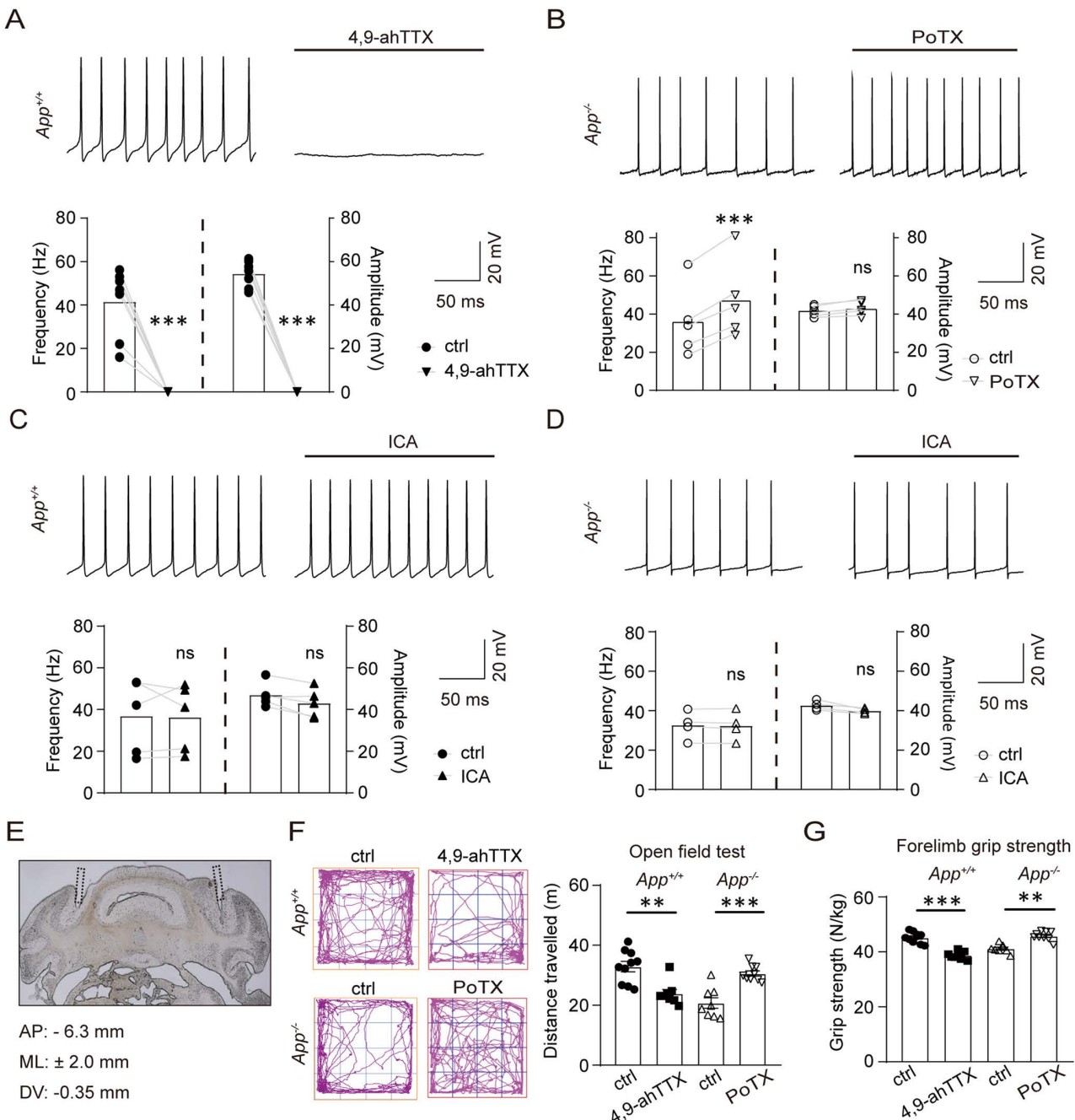

**Fig 5. Nav1.6 mediated the deficits of PC firing and motor function in APP-null mice. (A–D)** Electrophysiological characterization of PC spontaneous firing in acute cerebellar coronal slices (300-μm-thick). (A) Representative traces and quantification of spontaneous firing rate and spike amplitude before/during application of 200 nM 4,9-ahTTX in $App^{+/+}$ mice. $n = 7$ cells. (B) Representative traces and quantification of spontaneous firing rate and spike amplitude before/during application of 30 nM PoTX (Nav1.6 positive allosteric modulator) in $App^{-/-}$ mice. $n = 5$ cells/group. (C, D) Representative traces and quantification of spontaneous firing rate and spike amplitude before/during application of 350 nM ICA (Nav1.1 inhibitor) in $App^{+/+}$ (C) or $App^{-/-}$ (D) mice. Statistics: Paired $t$ test; * $P < 0.05$; ** $P < 0.01$; *** $P < 0.001$ vs. baseline; ns, non-significant. $n = 4$ cells/group. Scale bars are indicated in the images. **(E)** Morphology verification of cerebellar cannula placement in a coronal section of mice cerebellum. **(F)** Open field locomotion: Total distance traveled (m) in $App^{+/+}$ (4,9-ahTTX/vehicle) and $App^{-/-}$ (PoTX/vehicle) mice. Mice received bilateral microinjections of saline (vehicle), 200 nM 4,9-ahTTX, or 20 pM PoTX into the cerebellar cortex 20 min prior to testing. **(G)** Forelimb grip strength: maximal force/body weight in same cohorts as (F). Statistics (F, G): $App^{+/+}$+vehicle, $n = 10$; $App^{+/+}$+4,9-ahTTX, $n = 8$; $App^{-/-}$+vehicle, $n = 8$; $App^{-/-}$+PoTX, $n = 9$; One-way ANOVA with Tukey post-hoc; * $P < 0.05$; ** $P < 0.01$, *** $P < 0.001$; ns, non-significant. ctrl, control; 4,9-ahTTX, 4,9-anhydrotetrodotoxin; ICA, ICA-121431; PoTX, poneratoxin. The data underlying this Figure can be found in S1 Data.

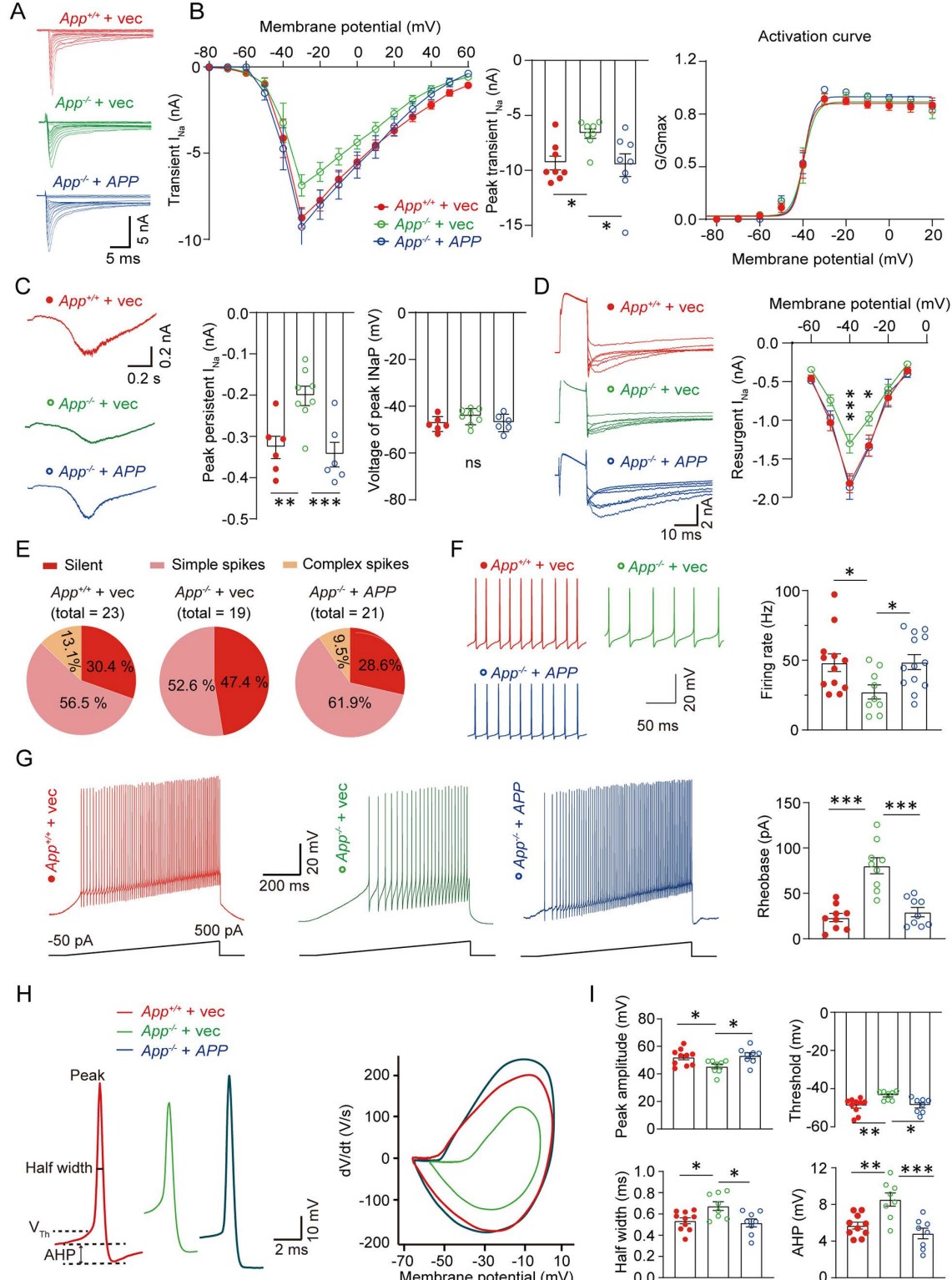

**Fig 6. Exogenous APP expression restored the Nav1.6 currents and ameliorated the abnormalities in the firing patterns of PCs of APP-null mice. (A, B)** Representative traces for whole-cell recording of (A) transient Na$^+$ current and (B) statistics of whole-cell *I–V* curve, peak transient currents, and activation curves in PCs of *App*$^{+/+}$+vec, *App*$^{-/-}$+vec, and *App*$^{-/-}$+APP mice. *N* = 8 cells for each group. **(C)** Representative persistent Na$^+$

current recorded in PCs. The right panels show statistics of the peak persistent currents and the voltage eliciting the peak currents. $App^{+/+}$+vec, $n=6$; $App^{-/-}$+vec, $n=8$; $App^{-/-}$+APP, $n=6$ cells. **(D)** Representative resurgent sodium currents and *I–V* curves recorded in PCs. $App^{+/+}$+vec, $n=8$; $App^{-/-}$+vec, $n=8$; $App^{-/-}$+APP, $n=8$ cells. **(E)** Pie chart showing the percentage of PC firing patterns in $App^{+/+}$+vec, $App^{-/-}$+vec, and $App^{-/-}$+APP mice. **(F)** Representative spontaneous simple spikes recorded in PCs. Quantification of firing rates analyzed from 1-min recordings during stable whole-cell access (≥10 min after breakthrough). $App^{+/+}$+vec, $n=12$; $App^{-/-}$+vec, $n=9$; $App^{-/-}$+APP, $n=13$ cells. **(G)** Representative membrane potential responses to depolarizing ramp current injections (−50 to +500 pA over 1 s) recorded from PCs under whole-cell current-clamp. Right: Rheobase quantification. $N=9$ cells/group. **(H)** Representative images of spontaneous action potentials and the *dV/dt* curves. **(I)** Statistics of peak amplitude, threshold, half-width, and afterhyperpolarization potential (AHP) of spontaneous action potentials shown in (H), $App^{+/+}$+vec, $n=10$; $App^{-/-}$+vec, $n=8$; $App^{-/-}$-+APP, $n=8$ cells. Scale bars are indicated in the images. One-way ANOVA and post-hoc Turkey's test. * $P<0.05$; ** $P<0.01$; *** $P<0.001$. The data underlying this Figure can be found in S1 Data.

## Exogenous APP expression in PCs restored Nav1.6-dependent GABAergic inputs to DCN neurons in APP-null mice

PC projections are the sole output of the cerebellar cortex and represent the major input to DCN neurons [29], forming approximately 70%–80% of all synaptic connections onto DCN cells [30,31]. To explore whether inhibitory transmission from PCs to DCN neurons was affected by APP deficiency, we performed whole-cell patch clamp recordings in cerebellar slices. DCN neurons innervated by PCs were labeled using *trans*-synaptic labeling, achieved by injecting AAV2/1-hSyn-CRE into the cerebellar cortex and AAV2/9-hSyn-DIO-EGFP in the DCN (Fig 7A). After 3 weeks of viral expression, EGFP-positive DCN neurons were recorded in brain slice (Fig 7B). No significant difference was observed in the number of EGFP-labeled DCN neurons between the different virus-injected groups. The amplitude and the frequency of spontaneous inhibitory post-synaptic currents (sIPSCs) in APP-null DCN neurons were lower than those in WT DCN neurons, and the abnormal GABAergic synaptic transmission APP-null mice was restored by APP expression in PCs (Fig 7C).

To further examine the role of Nav1.6 in AP transduction, we performed sIPSCs recordings during perfusion with 4,9 ah-TTX. This Nav1.6 blocker reduced the amplitude and the frequency of sIPSCs in all groups (Fig 7D). Importantly, application of 4,9 ah-TTX almost completely abolished the rescue effect of APP expression in GABAergic transmission of PC-DCN neurons (Fig 7D). These findings indicate that APP plays a crucial role in maintaining the inhibitory synaptic transmission from PCs to DCN neurons, and this function was Nav1.6-dependent. In addition, mIPSCs were also analyzed to identify whether APP deficiency influenced the AP firing-independent synaptic transmission. No significant difference in the amplitude or the frequency of miniature inhibitory post-synaptic currents (mIPSC) was observed among WT, APP-null+vec, and APP-null+*APP* groups (Fig 7E).

Taken together, our data provide strong evidence that APP was essential for sustaining the repetitive firing activity of PCs and for maintaining inhibitory synaptic transmission from PCs to DCN neurons via Nav1.6, a process critical for cerebellar-dependent motor performance.

## Discussion

Our results demonstrate that the APP deficiency leads to aberrant sodium currents, altered firing patterns in PCs, reduced inhibitory synaptic transmission to DCN, and impaired motor function mediated by PCs. Specifically, we show that the reduction in Nav1.6 sodium currents is a key factor contributing to the motor deficits observed in APP-null mice. These defects were recapitulated by PC-specific APP knockdown and rescued upon exogenous expression of APP in PCs. Taken together, our findings reveal a novel role for APP in motor control and provide new insights into the molecular mechanisms underlying this process.

Motor deficits in grip strength and locomotion in APP-deficient mice have been reported by several groups in the past [8,32]. However, the underlying mechanisms have remained poorly understood. The absence of prominent learning and memory phenotypes in single APP knockout mice has been attributed to functional compensation by APP family members, APLP1 and APLP2, with more pronounced deficits observed in APP and APLP double knockout mice [10]. Interestingly,

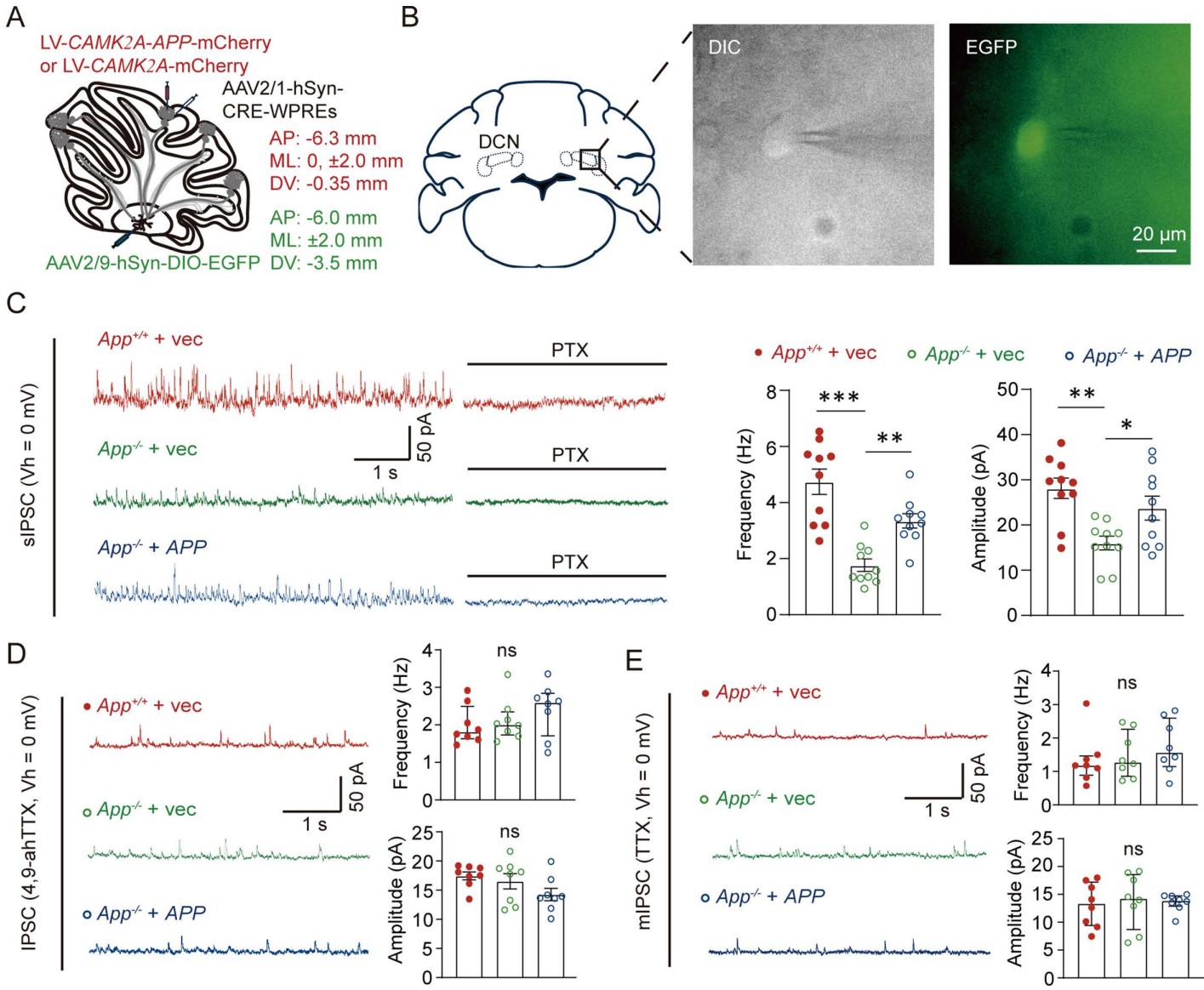

**Fig 7. Exogenous amyloid precursor protein (APP) expression restored the defects of the sIPSCs evoked by Nav1.6-mediated Purkenji cell inputs in APP-null mice. (A)** Schematic of the viral injection strategy to achieve APP overexpression in PCs and *trans*-synaptic tracing of downstream neurons in the DCN. **(B)** Representative example of fluorescence-assisted selection of cells for patch-clamp recordings. Images from an acute cerebellar slice observed in differential interference contrast (DIC, left), or fluorescence mode (EGFP, right). **(C)** Left panel: Representative traces of sIPSCs in the PC-projected DCN neurons held at 0 mV. Brain slices of $App^{+/+}$ + vec, $App^{-/-}$ + vec and $App^{-/-}$ + APP mice were recorded; GABA$_B$-receptor antagonist picrotoxin (PTX, 100 μM) was perfused at the end of the recordings to validate IPSCs. Right panel: statistics of IPSCs frequency and amplitude, $n = 10$ cells for each group. **(D)** IPSCs recorded from the PC-projected DCN neurons with 4,9-ahTTX (200 nM), $n = 8$ cells for each group. **(E)** Miniature IPSCs (mIPSCs) recorded from PC-projected DCN neurons in the presence of TTX (1 μM), $n = 8$ for each group. Scale bars as presented in the images. A 5-min recording segment from a stable period, defined as commencing at least 10 min after obtaining whole-cell access, was used for statistical analysis. Data are represented as the mean with SEM (C, D IPSCs amplitude) or the median with interquartile range (D IPSCs frequency, E). One-way ANOVA and Kruskal–Wallis test. * $P < 0.05$; ** $P < 0.01$; *** $P < 0.001$; ns, non-significant. The data underlying this Figure can be found in S1 Data.

APLPs do not compensate for certain phenotypes seen in APP knockout mice, particularly those related to motor function and body size, suggesting that APP plays a unique role in the regulation of motor functions within the APP family. A key question remains: in which brain regions or cell types does APP mediate its effects on motor control? Within the motor system, APP knockout mice exhibit normal neuromuscular junction morphology [11]. Furthermore, mice with conditional APP knockout targeted to excitatory neurons via the Nex promoter show intact motor functions [7]. These observations guided our focus on cerebellar PCs to investigate APP's role in motor control. Our exogenous rescue and conditional knockdown experiments demonstrate that APP in PCs is critical for motor function, appropriate expression of APP in PCs is essential for normal motor performance, particularly specific behaviors such as locomotion and forelimb grasping ability. However, it should be noted that this study focuses solely on cerebellar PCs; we cannot exclude the possibility that APP also modulates motor function via other regions of the cerebrospinal circuit, such as spinal motor neurons. Nav1.6-mediated persistent sodium current ($I_{NaP}$) contributes to postural tone and amplifies locomotor outputs in spinal cord bistable motor neurons [33]. Given that APP is also expressed in spinal motor neurons [34], it is likely that spinal APP may similarly participate in the regulation of motor function.

As a key component in the formation of internal models of the cerebellum, PCs have the capacity to process a vast array of signals [35]. The output information from PCs is encoded by two distinct types of APs: high-frequency SSs and low-frequency CSs. A single PC receives excitatory input from between 100,000 and 200,000 parallel fibers, which modulate the intrinsically driven high-frequency SS discharge [18,36]. In monkeys, increased SS discharge during grasping and lifting movements was observed in 56% of PCs [37], supporting the hypothesis that the cerebellum plans or controls movements through a forward internal model. Complex spikes are characterized by a large Na$^+$ somatic spike followed by a burst of smaller spikelets, which are generated by a massive depolarization of the entire PC induced by climbing fiber activation [35]. These CSs serve as error signals crucial for online correction and motor learning [38]. Although the precise representations of PC firing patterns remain incompletely understood, there is a clear correlation between PC firing and motor output. Therefore, aberrant firing patterns in PCs are likely to influence both grip force and locomotion. Specifically, the decline in SS firing rates and the reduction in CS generation may contribute to the motor impairments, such as the deficits in grip strength and locomotion observed in APP-null mice.

We previously reported that APP interacts with Nav1.6 and regulates its function in HEK293 cells by enhancing its cell surface distribution [21]. Nav1.6 is critical to the electrophysiological activity of PCs, comprising 82%–92% of resurgent sodium currents, 69% of persistent sodium currents, and more than 37% of the transient sodium currents [18]. Previous studies have shown that Nav1.6 and Nav1.1 are two major sodium channel subtypes expressed in PCs. We employed selective blockers/inhibitors combined with eliciting protocols to isolate different sodium current components produced by Nav1.1 or Nav1.6. Our data indicate that APP regulates Nav1.6 rather than Nav1.1 currents in transient $I_{Na}$. APP regulates the persistent and resurgent $I_{Na}$, which is predominantly produced by Nav1.6. Thus, APP plays a crucial role in the AP firings of PCs by modulating Nav1.6 currents. We further demonstrated that PoTX, a positive allosteric modulator of Nav1.6 that reduces channel inactivation, consequently enhanced the persistent $I_{Na}$—, key determinant of repetitive AP firing. This PoTX-mediated enhancement likely compensates for the functional deficits resulting from reduced surface distribution of Nav1.6 in APP-null PCs, as evidenced by the restoration of firing properties and amelioration of motor deficits. These results further support the conclusion that Nav1.6 serves as the primary effector mediating APP-dependent regulation of cerebellar PC function. However, as shown in Fig 2, APP-null mice exhibited a higher threshold for AP generation and an enhanced AHP. These electrophysiological alterations may collectively contribute to the observed reduction in spontaneous firing frequency of PCs. In addition to regulating Nav1.6, APP may also modulate the threshold for AP generation and AHP through other mechanisms. Additionally, Nav1.6 mediates the saltatory propagation of APs, reflected by the output of PCs. APP-null mice exhibited abnormal synaptic output to post-synaptic DCN neurons, contributing to impaired motor functions. Notably, the motor deficiencies observed in APP-null mice differ from those in Nav1.6-knockout mice (such as *med* mice). This discrepancy may be explained by the fact that Nav1.6 is only one of the

downstream targets of APP, and the partial modulation of Nav1.6 by APP is not as dramatic as the complete knockout of the *Scn8a* gene.

In summary, this study builds on previous findings that APP-null mice exhibit motor function deficits and identifies a novel mechanism through which APP regulates motor control via Nav1.6 in PCs.

## Materials and methods

### Ethics statement

All mouse maintenance and experimental procedures were approved by the Institutional Animal Care and Use Committee and the Office of Laboratory Animal Resources of Xuzhou Medical University (protocol no. 202207S009). All procedures were conducted in accordance with the Regulations for the Administration of Affairs Concerning Experimental Animals (2017) in China and the United States Public Health Service Policy on Humane Care and Use of Laboratory Animals.

### Animals

The APP-null mice were as described previously [8]. C57BL/6J background APP-null mice were purchased from the Jackson Laboratory (stock no. 004142). C57BL/6J mice were purchased from GemPharmatech (Nanjing, China). The mice were housed in groups, with a maximum of five individuals per cage, under a 12 h light/dark cycle. Male mice were used for behavioral tests to eliminate the influence of the estrous cycle, and both sexes were used in other experiments. All mice used in the experiments were 2–3-month-old, unless otherwise specified. Mice were randomly assigned to control and treatment groups by blindly selecting from the same cage of littermates.

### Antibodies and plasmids

The antibodies used in this study included: rabbit anti-Nav1.6 (*Scn8a*) (IgG, 1:300, Alomone Labs, Catalogue No. ASC-009), mouse anti-GFP (1:1,000, Roche Applied Science, Catalogue No. 11814460001), anti-flag-HRP (1:10000, AlpVHHs, Catalogue No. 016-303-005). The plasmids used in this study included: pcDNA3.1-SP-Flag-hAPP695 was a gift from Dr. Gavin S. Dawe, pcDNA3.1-SP-Flag-hAPP (18-612AA)and pcDNA3.1-hAPP-flag (637-695AA)were constructed by Genecfps (Wuxi, China). pCMV-SCN1A (human) -EGFP-Neo was purchased from MIAOLING PLASMID (P51606).

### Virus injection

Specific SP-Flag-APP (full-length of mice APP) sequence was incorporated into a recombinant lentivirus viral vector, which features a *CAMK2A* promoter to drive APP expression (LV-*CAMK2A*-SP-Flag-APP-3'UTR-IRES2-mCherry-WPRE). The LV-*CAMK2A*-mCherry was used as a control. Additionally, to investigate the roles of specific APP proteolytic fragments, lentiviral vectors expressing SP-Flag-sAPPα (LV-*CAMK2A*-SP-Flag-APP(AA18-612)-3'UTR-IRES2-mCherry-WPRE) and AICD-Flag (LV-*CAMK2A*-APP(AA637-695)-Flag-3'UTR-IRES2-mCherry-WPRE) were generated. The LV-*CAMK2A*-MCS-IRES2-mCherry vector was used as the control. The standard titers of LVs were ≥1 × 10^8 TU/ml.

To achieve cell-type-specific knockdown of APP in cerebellar PCs, we employed a dual recombinant adeno-associated virus (rAAV) system [23,24]. A mixture (1:1 ratio) of rAAV-L7-CRE-P2A-mCherry-WPRE-hGH (utilizing the PC-specific L7 promoter to express Cre recombinase) and rAAV-CMV-DIO-(EGFP-U6)- *App* shRNA (expressing APP-targeting shRNA conditionally upon Cre-mediated recombination) was injected. A mixture of rAAV-L7-CRE-P2A-mCherry-WPRE-hGH and rAAV-CMV-DIO-(EGFP-U6)-*App* shRNA (scrambled sequence served as control). *App* shRNA sequence: 5′-GCACT AACTTGCACGACTATG-3′; Scrambled *App* shRNA: 5′-GGTCCAAACCGTCCAGTTAAT-3′. AAV2/1-hSyn-CRE-WPREs, and AAV2/9-hSyn-DIO-EGFP were purchased from BrainVTA Biotechnology (Wuhan, China). The viral titers were 5 × 10^12 TU/ml. All viruses were aliquoted and stored at −80 °C until use.

The methods for stereotaxic surgery were described before [39,40]. Briefly, mice were anesthetized with ketamine (100 mg/kg, i.p., 210220BL, Hengrui, China) and xylazine (10 mg/kg, i.p., YXK18801, Yingxin LAB, China). The virus was injected using a syringe nanoliter infusion/withdraw pump (KW-ZSB, KEW BASIS) attached to a 1 µl Hamilton syringe at a rate of 0.1 µl/min. For exogenous expression in PCs, 300 nl of the respective lentivirus (holo-APP, sAPPα, AICD, or control) was injected into the cerebellar cortex (coordinates, bregma: $AP = -6.3$ mm; $ML = \pm 2.0$ mm; $DV = -0.35$ mm). For APP knockdown in PCs, 200 nl of the virus mixture was injected into the cerebellar cortex at the same coordinates used for lentiviral APP expression (bregma: $AP = -6.3$ mm; $ML = 0, \pm 2.0$ mm; $DV = -0.35$ mm). For the PC-DCN *trans*-synaptic tracing, we injected 200 nl AAV2/1-hSyn-CRE-WPREs into cerebellar cortex and 200 nl AAV2/9-hSyn-DIO-EGFP into DCN (coordinates, bregma: $AP = -6.0$ mm; $ML = \pm 2.0$ mm; $DV = -3.5$ mm). After surgery, mice were returned to their home cages and allowed to recover for at least three weeks before further experiments.

## Intracerebellar drug infusion

Mice were anesthetized with ketamine (100 mg/kg, i.p.) and xylazine (10 mg/kg, i.p.) prior to stereotaxic surgery. Bilateral sterile guide cannula (o.d.: 0.36 mm; i.d.: 0.3 mm, AOGUAN Biotechnology) were implanted targeting the cerebellar cortex (bregma coordinates: $AP = -6.3$ mm, $ML = \pm 2.0$ mm; $DV = -0.35$ mm) for microinjection. Following a minimum recovery period of 4 days, mice received microinjections of either saline (0.5 µl/lateral), the Nav1.6 blocker 4,9-ahTTX (200 nM, 0.5 µl/lateral; MCE), or the Nav1.6 positive allosteric modulator PoTX (20 pM; 0.5 µl/lateral, MCE, HY-P10234A) 20 min before behavioral testing. Drug infusion into the cerebellar cortex was performed over 5 min using blunted microliter syringes (Hamilton, 1 µl, 25 gauge), extending 0.2 mm beyond the guide cannula tip. The injection needles remained in place for an additional 2 min post-infusion.

## Laser capture microdissection (LCM) and RT-qPCR

The LCM system (ZEISS, PALM MicroBeam) was preheated before use. After anesthetization with ketamine and xylazine, mice were rapidly decapitated. The extracted brains were immediately embedded in optimal cutting temperature compound (OCT, Neg-50, Epredia, USA) and sectioned into 15-µm-thick slices using a cryostat. The tissue sections were mounted onto clean glass slides and positioned within an Axio Observer inverted microscope (ZEISS). The LCM system parameters were set according to the manufacturer's instructions: 355-nm pulse laser, 55-µJ pulse energy, and 100% cutting speed. Following identification of fluorescence-positive PCs based on characteristic morphology (large flask-shaped somata, 20–30 µm diameter), targeted cells were collected via ultraviolet laser pressure catapulting using inverted-beam geometry for anti-gravitational ejection into adhesive cap tubes (AdhesiveCap 500, Zeiss). The harvested cells were transferred to RNase-free Eppendorf tubes for mRNA extraction and quantitative PCR (qPCR) analysis. The detailed qPCR protocols and quantification methods were previously described [39]. The primer sequences used for mRNA quantification: holo-APP and sAPPα (Forward: GAAGCCATGCTCAATGACC; Reverse: ATGCTTTAGGGTGTGCTGTC); *Gapdh* (Forward: ATGGTGAAGGTCGGTGTGAACG; Reverse: CGCTCCTGGAAGATGGTGATGG); AICD (Forward: ATCATG-GTGTGGTGGAGGTTG; Reverse: AGGTTGGATTTTCGTAGCCGT); *Scn8a* (Forward: AGGCCCCGACAGTTTCAAG; Reverse: GGGTGGTTTCTTGAGCTTGC). The relative quantification was calculated by $2^{-\Delta\Delta ct}$ method.

## Whole-cell patch-clamp recording

Mice were anesthetized with isoflurane (3%–4%, R510-22, RWD, China) and rapidly decapitated. The cerebellum was carefully dissected from the extracted brain using a sharp blade. Coronal cerebellar slices (300-µm-thick) were prepared using a vibratome (VT1200S, Leica Microsystems, Nussloch, Germany) in ice-cold sucrose-based artificial cerebrospinal fluid (sACSF) (sucrose 212, 3 mM KCl, 1.25 mM $NaH_2PO_4$, 26 mM $NaHCO_3$, 10 mM glucose, 7 mM $MgCl_2$, PH 7.3, 320 mOsm), equilibrated with 95% $O_2$ and 5% $CO_2$. Slices were allowed to recover in ACSF at 32 °C for 45–60 min, and

then incubated at room temperature (22–24°C) in normal ACSF (124 mM NaCl, 2.5 mM KCl, 1.25 mM $NaH_2PO_4$, 1.3 mM $MgSO_4$, 2 mM $CaCl_2$, 26 mM $NaHCO_3$, and 20 mM glucose; titrated to pH 7.4 with NaOH) for at least 30 min before use. PCs in brain slices were visualized under an upright microscope (Olympus BX51WI) equipped with an infrared CCD camera. Whole-cell patch-clamp recordings were performed using a MultiClamp 700B amplifier (Axon Instruments), a Digidata 1550B analog-to-digital converter (Axon Instruments), and pClamp 10.7 software (Molecular Devices, San Jose, CA). Patch electrodes had a resistance of 2–4 MΩ when filled with either a firing-recording internal solution (140 mM K-methylsulfate, 7 mM KCl, 2 mM $MgCl_2$, 10 mM HEPES, 0.1 mM EGTA, 4 mM $Na_2$-ATP, 0.4 mM GTP-Tris) or a low-chloride internal solution for inhibitory synaptic transmission recording (135 mM K-gluconate, 5 mM KCl, 0.2 mM EGTA, 0.5 mM $CaCl_2$, 10 mM HEPES, 2 mM Mg-ATP, 0.1 mM GTP). The pH was adjusted to 7.2 using Tris-base, and the osmolarity was adjusted to 300 mOsm with sucrose.

ICA121431 (350 nM, MCE, HY-16787) or 4,9-ahTTX (200 nM, GlpBio, GC42327) were dissolved in ACSF to selectively inhibit Nav1.1 or Nav1.6, respectively. For recording mIPSCs in PCs, 1 μM TTX (L1808N, Puhuashi Technology, China) was dissolved in ACSF. IPSCs were recorded at a holding potential of 0 mV. For whole-cell sodium current recordings, 0.2 mM $CdCl_2$ (202908, Sigma, USA) was added to block calcium channels, and 140 mM TEACl (T2265, Sigma, USA) was included to inhibit potassium channels. Patch electrodes were filled with an intrapipette solution containing 108 mM CsF, 6 mM $MgCl_2$, 1.8 mM EGTA, 10 mM HEPES, 4 mM Na2-ATP, and 0.3 mM Tris-GTP, pH 7.3, 280–290 mOsm. Transient sodium currents were elicited by a series of 10 mV steps from a holding potential of −90 to +60 mV. Persistent current was elicited by a slow ramp increase from −90 to +30 mV at a rate of 0.12 mV/ms. Resurgent sodium current was elicited following a step to +30 mV (from a holding potential of −90 mV) by a series of 10-mV-depolarizing voltage steps from −60 to +10 mV. Sodium currents were verified by their complete block during bath application of 1 μM TTX in ACSF. $I_{Nav1.6}$ was isolated by subtracting the sodium currents recorded during bath application of 200 nM 4,9-ahTTX from those recorded before the blocker application. $I_{Nav1.1}$ was isolated by subtracting the sodium currents recorded during bath application of 350 nM ICA-121431 from those recorded before the inhibitor application.

All experiments were conducted to collect data during a stable period, which was defined as at least 10 min after establishing whole-cell access. Electrophysiological data were analyzed offline using Clampfit 11.2 software (Molecular Devices). To ensure high-quality intracellular recordings, only cells exhibiting a stable resting membrane potential and access resistance with no more than 20% variation were considered valid for analysis.

## Behavioral tests

**Open field test.**  Each mouse was placed in the center of an open field arena (a cube with 50 cm long sides and a height of 50 cm) and allowed to explore freely for 10 min. Locomotor activity was recorded using a video camera controlled by ANY-Maze 14.0 software. The total travel distance of each mouse within the arena was recorded and analyzed.

**Grip strength test.**  The grip strength of mouse forelimbs was assessed using a grid connected to a strength sensor (YLK-2N, ELECALL, China), as previously described [41]. Mice were allowed to grip a metal grid with their forelimbs, after which they were lifted by the tail and gently pulled backward until they released the grid. During the test, the hind limbs were kept away from the grid. Grip strength was measured 10 times, and the mean of the top five values was used for analysis. All grip strength values were normalized to body weight.

**Rotarod test.**  Mice were trained to run at an accelerating speed (from 4 to 40 rpm, with an acceleration of 0.1 rpm/s) on a rotarod instrument (ZH-600B, ZhenghuaBiologic, China). Animals underwent three trials of accelerating rotarod running, with a time limit of 6 min per trial. The latency to fall was recorded as an indicator of training performance. A 6-min rest period was provided between trials to minimize stress and fatigue.

**Footprint assay.**  This assay was used to assess gait abnormality. A recording paper (100 cm × 10 cm) was placed at the bottom of a clear plexiglass tunnel (100 cm × 10 cm × 10 cm), with a darkened cage at the end of the tunnel [42]. Mice with different ink-painted front and rear paws were allowed to travel through the tunnel. The following gait parameters

were then measured using the footprints on the recording paper: (a) stride length, the distance between two successive rear paw prints on the right side; (b) stance length, the distance between the left and right rear paws; (c) sway length, the vertical distance between the left and right rear paws; and (d) overlap length, the distance between the center of the front and rear paw prints on the right side (S1D Fig).

**Balance beam test.** Mice were assessed for motor coordination using a horizontal round beam (100 cm length × 6/12 mm diameter, wooden) elevated 30 cm above the floor. The beam terminated at a darkened escape platform (20 × 20 cm). After two consecutive days of training (3 trials/day per beam width; data not recorded), formal testing on day 3 was recorded: (a) traversal latency (from beam entry to platform access) and (b) total limb slips (counted by blinded observers from video recordings). Three trials per beam width were averaged for analysis.

## Immunofluorescence

Mice were anesthetized with ketamine and xylazine (100 mg/kg and 10 mg/kg, i.p.), then subjected to transcardial perfusion with 20 ml phosphate-buffered saline (PBS), followed by 20 ml 4% paraformaldehyde (PFA). Brains were extracted and post-fixed in 4% PFA overnight at 4 °C. Following fixation, tissues were dehydrated stepwise in 15% and 30% sucrose and embedded in OCT for sectioning. Cerebellar sections (20-μm-thick) were cut using a cryostat (Leica) and mounted on gelatin-coated slides (Citoglas). Sections were rinsed with PBS and subsequently incubated overnight at 4 °C with primary antibodies diluted in PBS containing 0.1% Triton X-100, 0.05% Tween-20, and 1% goat serum. Rabbit anti-APP IgG (1:500, Abcam, Catalogue No. AB32136), mouse anti-Calbindin-D28K IgG (1:200, Proteintech, Catalogue No. AB2881769), rabbit anti-Na$_v$1.6 (*Scn8a*) IgG (1:300, Alomone Labs, Catalogue No. ASC-009) were used in the study. Sections were washed three times (10 min each) with PBS, then incubated with Alexa 488- or Alexa 594-conjugated secondary antibodies (1:2,000; Jackson ImmunoResearch) for 1 h at room temperature, protected from light. Finally, images were captured using a fluorescence microscope (Olympus IX81) controlled by Cellsens Standard software (Olympus, Japan), and processed using ImageJ (NIH, Bethesda, MD, USA).

## Golgi staining

Golgi-Cox staining was performed using the Golgi-Cox OptimStain PreKit (PK401, HiTO, USA) according to the manufacturer's instructions. Briefly, the solution A and solution B from the kit were mixed and added to the light-protected glass bottle 24 h in advance. The following day, brains were immersed in the solution A+B solution and stored in the dark at room temperature for 2 weeks, with the impregnation solution replaced after 24 h. Afterward, the brains were transferred to solution C and stored in the dark at room temperature for 3–7 days. Brain tissue was then sectioned into 120-μm-thick slices using a vibratome and mounted onto gelatin-coated slides. Excess water was removed, and a small amount of solution C was added dropwise. After 2 min, the slide was tilted to dry. Slides were rinsed three times in ddH$_2$O and stained for 10 min using a freshly prepared staining solution consisting of one part solution D, one part solution E, and two parts ddH$_2$O. The slides were then washed in ddH$_2$O for 10 min. Sections were dehydrated and cleared in xylene (10023418, Sinopharm, China) and mounted with neutral gum (G8590, Solarbio, China). Images were obtained using an Olympus BX53 microscope with Olympus CellSens Standard software and analyzed by ImageJ Fiji. Dendritic spine density analysis was conducted by counting the dendritic spines along the full length of the apical dendrite of each PC. Twelve cells were counted for statistical analysis by a researcher blinded to the treatment group. Sholl analysis was performed using ImageJ software. The number of intersections between the circles and dendrites were plotted against the radii. The diameter of the primary dendrite was measured at 10 μm from the soma.

## Cell culture and transfection

The HEK293 cell line stably expressing Nav1.6 was described previously [21]. These cells were maintained in DMEM supplemented with 10% (v/v) fetal bovine serum (FBS) and 200 μg/ml G418 (Absin). Normal HEK293 cells were cultured in

DMEM containing 10% FBS and 1% penicillin/streptomycin. Plasmid transfections used Lipofectamine 2000 (GlpBio) per manufacturer protocol. Cells were harvested 48 h post-transfection for downstream assays.

## Co-IP assay

Transfected HEK293 cells were harvested and lysed in ice-cold lysis buffer (150 mM NaCl, 30 mM HEPES, 10 mM NaF, 1% (v/v) Triton X-100, 0.01% (w/v) SDS, and complete protease inhibitor mixtures, pH 7.5). The lysates were rotated for 2 h at 4 °C and centrifuged at 12,000 rpm for 20 min. The supernatants were collected and incubated overnight at 4 °C with Anti-Flag Affinity Beads (smart-lifesciences, Catalogue No. SA042001), then washed three times with ice-cold lysis buffer. Finally, the samples were boiled in SDS loading buffer and analyzed by Western blotting.

## Statistical analysis

Data were analyzed using GraphPad Prism 8.0. Normality was assessed with the Shapiro–Wilk test. Normally distributed data are expressed as mean with SEM, while non-normally distributed data are presented as median with interquartile range. Outliers were identified using Dixon's $Q$-test at the 95% confidence level. For comparisons between two groups, unpaired or paired Student's t-tests were used, as appropriate. Comparisons across three or more groups were performed using one-way ANOVA (for normally distributed data with homogeneous variance) or the Kruskal–Wallis test (for non-normal distributions), followed by appropriate post hoc tests. A $P$-value $< 0.05$ was considered statistically significant.

## Declaration of generative AI and AI-assisted technologies in the writing process

During the preparation of this work, the authors used ChatGPT and DeepSeek in order to improve language and readability. After using this tool/service, the authors reviewed and edited the content as needed and take full responsibility for the content of the publication.

## Supporting information

**S1 Fig. APP-null mice exhibit motor function deficits.** A battery of motor function tests was performed in $App^{+/+}$ (wild-type) and $App^{-/-}$ mice, $n = 10$ for each group. **(A)** Open field test: representative trajectory plots and statistics of the distance traveled in the open field. **(B)** Statistics of grip strength test. **(C)** Statistics of rotarod test. **(D)** Representative trajectory maps and schematic measurement of footprint. **(E)** Statistics of sway length, stride length, stance length and overlap of the footprint assay in $App^{+/+}$ and $App^{-/-}$ mice. **(F)** Statistics of mouse body weight. **(G)** Balance beam test. The time taken for mice to traverse beams of two different widths (6 and 12 mm) and the number of slips were recorded. Bar-charts show the quantification of traversal time and number of slips per mouse. $N = 10$ for each group. Scale bars are indicated in the images. Student $t$ test: * $P < 0.05$; ** $P < 0.01$; *** $P < 0.001$; ns, non-significant. The data underlying this Figure can be found in S1 Data.
(TIF)

**S2 Fig. L7 promoter-driven Cre recombinase expression is restricted to PCs in the cerebellar cortex.** Representative sagittal cerebellar sections (20 μm) from wild-type mice injected with rAAV-L7-CRE-P2A-mCherry-WPRE-hGH, showing mCherry (Cre reporter, red) and immunofluorescence for the PC marker calbindin (green). Merged images reveal exclusive co-localization within PCs, confirming recombinant expression specificity. Scale bars: 100 μm (main); 20 μm (insets).
(TIF)

**S3 Fig. APP deficiency does not alter PC development or induce degeneration. (A–C)** Analyses in 2-month-old and **(D–F)** 12-month-old $App^{-/-}$ versus $App^{+/+}$ mice. (A, D) Representative sagittal sections of the cerebellar cortex showing

Golgi-stained PCs, with corresponding Sholl analysis of dendritic arborization. Concentric circles (radius increment: 25 µm) are overlaid on the binary-traced PC image to quantify intersections with the dendritic processes. $App^{+/+}$ (2-month-old), $n = 13$; A$pp^{-/-}$ (2-month-old), $n = 12$; $App^{+/+}$ (12-month-old), $n = 10$; $App^{-/-}$ (12-month-old), $n = 10$. (B, E) Measurement of primary dendrite diameter at a distance of 10 µm from the soma. $App^{+/+}$ (2-month-old), $n = 15$; $App^{-/-}$ (2-month-old), $n = 19$; $App^{+/+}$ (12-month-old), $n = 12$; $App^{-/-}$ (12-month-old), $n = 12$. (C, F) Quantification of dendritic spine density on distal dendrites of PCs. Red arrowheads indicate spines along a distal dendrite. $App^{+/+}$ (2-month-old), $n = 14$; $App^{-/-}$ (2-month-old), $n = 14$; $App^{+/+}$ (12-month-old), $n = 15$; $App^{-/-}$ (12-month-old), $n = 15$. Scale bars are indicated in the images. Data are from 4 mice per group; Student $t$ test, ns: not significant. The data underlying this Figure can be found in S1 Data.
(TIF)

**S4 Fig. Validation of Nav1.6 and Nav1.1 contributions to sodium currents in PCs.** Whole-cell voltage-clamp recordings from wild-type ($App^{+/+}$) PCs characterizing voltage-gated sodium current subtypes. **(A, D)** Transient current ($I_{NaT}$): elicited by depolarizing steps from −90 to +60 mV. **(B, E)** Persistent current ($I_{NaP}$): evoked using a 1,000-ms voltage ramp from −90 to +30 mV. **(C, F)** Resurgent current ($I_{NaR}$): induced following a +30 mV step (from $V_{hold} = −90$ mV) with repolarizing steps from −60 to +10 mV (10 mV increments). (A–C) Current traces before (black) and during application of 1 µM tetrodotoxin (TTX, pan-NaV blocker; red). (D–F) Current traces before (black) and during co-application (orange) of 200 nM 4,9-anhydrotetrodotoxin (Nav1.6 blocker) and 350 nM ICA-121431 (Nav1.1 inhibitor). Scale bars are shown in the figure.
(TIF)

**S5 Fig. Nav1.6 Mediates most of the persistent and resurgent sodium currents in cerebellar PCs. (A)** Persistent Na$^+$ currents elicited by a 1,000-ms ramp from −90 to +30 mV. Representative traces and statistical bar-charts showing amplitude of persistent $I_{Na}$ before or during bath application of Nav1.6 specific blocker 4,9-ahTTX (200 nM), or Nav1.1 specific inhibitor ICA-121431 (ICA, 350 nM) in wild-type ($App^{+/+}$) mice. 4,9-ahTTX: $n = 6$ cells, ICA: $n = 6$ cells. **(B)** Resurgent Na$^+$ currents elicited following a step to +30 mV (from a holding potential of −90 mV) by a series of 10 mV depolarizing voltage steps from −60 to +10 mV. Representative traces and statistical bar-charts showing amplitude of resurgent $I_{Na}$ before or during bath application of 4,9-ahTTX or ICA in wild-type mice. 4,9-ahTTX: $n = 6$ cells, ICA: $n = 6$ cells. The Nav1.6 or Nav1.1 currents were determined by subtracting the sodium currents recorded during bath application of specific blockers/inhibitors from that before drug application. Scale bars are indicated in the figure. Statistics: Unpaired $t$ test; *** $P < 0.001$. The data underlying this Figure can be found in S1 Data.
(TIF)

**S6 Fig. APP deficiency did not affect the mRNA level of *Scn8a* in cerebellar PCs.** QPCR analysis of LCMed cerebellar PCs in $App^{+/+}$ or $App^{-/-}$ mice. Statistics: $App^{+/+}$, $n = 5$; $App^{-/-}$, n = 4. Unpaired $t$ test; ns, non-significant. The data underlying this Figure can be found in S1 Data.
(TIF)

**S7 Fig. Antibody validation for Nav1.6 and Nav1.1-EGFP in transfected HEK293 cells.** Total lysates of HEK293 cells expressing either Nav1.6 or Nav1.1-GFP were immunoblotted with Nav1.6 (Alomone Labs #ASC-009) or EGFP (Roche Applied Science, #11814460001) antibody.
(TIF)

**S8 Fig. PoTX potentiates persistent sodium currents in APP-null PCs.** Whole-cell voltage-clamp recordings of persistent sodium currents in $App^{-/-}$ PCs before (black) and during bath application of 30 nM PoTX (red). Currents were elicited by a 1000-ms ramp to +30 mV from a holding potential of −90 mV. Scale bars as indicated.
(TIF)

**S1 Raw Images. Raw images for blots.**
(TIF)

**S1 Data. Source data and statistical details for figures.**
(XLSX)

## Acknowledgments

The authors sincerely thank Dr. Gavin S. Dawe for generously providing the hAPP695 plasmid and the Nav1.6-expressing HEK293 cell line, as well as for his expertise in APP biology. We are also grateful to Dr. Lan Bao and Dr. You-Sheng Shu for their helpful discussions and invaluable insights into voltage-gated sodium channels.

## Author contributions

**Conceptualization:** Miao-Jin Ji, Chao Liu.

**Funding acquisition:** Miao-Jin Ji, Chao Liu.

**Investigation:** Miao-Jin Ji, Tong-Xuan Wu, Chenhao Tian, Xiang Cao, Ruyuan Wei, Jiao Yang, Chao Liu.

**Methodology:** Miao-Jin Ji, Tong-Xuan Wu, Chenhao Tian, Xiang Cao, Yin-Yin Yang, Xinran Meng, Huanyao Tang, Jiao Yang, Xin Tang, Chao Liu.

**Project administration:** Miao-Jin Ji, Chao Liu.

**Resources:** Chao Liu.

**Supervision:** Miao-Jin Ji, Chao Liu.

**Validation:** Miao-Jin Ji, Tong-Xuan Wu.

**Visualization:** Miao-Jin Ji, Tong-Xuan Wu, Chao Liu.

**Writing – original draft:** Miao-Jin Ji, Tong-Xuan Wu, Chao Liu.

**Writing – review & editing:** Miao-Jin Ji, Chenhao Tian, Xiang Cao, Tiantao Cui, Chao Liu.

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
