## [Editor Report · Decision Letter 0]

25 Feb 2025

Dear Dr Liu,

Please disregard the other email which you should also have received from us, where I accidentally clicked send too early.

Thank you for submitting your manuscript entitled "Amyloid Precursor Protein Modulates Cerebellar Purkinje Cell Activity and Motor Function via Regulation of Nav1.6 Currents" for consideration as a Research Article by PLOS Biology.

Your manuscript has now been evaluated by the PLOS Biology editorial staff and I am writing to let you know that we would like to send your submission out for external peer review.

Once your full submission is complete, your paper will undergo a series of checks in preparation for peer review. After your manuscript has passed the checks it will be sent out for review. To provide the metadata for your submission, please Login to Editorial Manager (https://www.editorialmanager.com/pbiology) within two working days, i.e. by Feb 27 2025 11:59PM.

Kind regards,

Taylor

Taylor Hart, PhD,

Associate Editor

PLOS Biology

thart@plos.org

---

## [Decision Letter · Decision Letter 1]

3 Apr 2025

Dear Dr Liu,

Thank you for your patience while your manuscript "Amyloid Precursor Protein Modulates Cerebellar Purkinje Cell Activity and Motor Function via Regulation of Nav1.6 Currents" was peer-reviewed at PLOS Biology. Your manuscript has been evaluated by the PLOS Biology editors, an Academic Editor with relevant expertise, and by several independent reviewers.

As you will see in the reviewer reports, which can be found at the end of this email, although the reviewers find the work potentially interesting, they have also raised a substantial number of important concerns. Based on their specific comments and following discussion with the Academic Editor, it is clear that a substantial amount of work would be required to meet the criteria for publication in PLOS Biology. However, given our and the reviewer interest in your study, we would be open to inviting a comprehensive revision of the study that thoroughly addresses all the reviewers' comments. Given the extent of revision that would be needed, we cannot make a decision about publication until we have seen the revised manuscript and your response to the reviewers' comments. Your revised manuscript would need to be seen by the reviewers again, but please note that we would not engage them unless their main concerns have been addressed.

The reviewers complimented the study design and the novelty of the results. However, R1 & R3 raised concerns over several missing controls, while R2 pointed out the absence of novel mechanistic insights. We believe that the study would be suitable for PLOS Biology after better substantiation of the major findings, and would like to invite you to carry out a Major Revision of your manuscript. The revised submission should include data from additional experiments, including further manipulations of APP/Nav1.6, as well as more detailed behavioral characterizations of the mutant mice. You should carefully consider the experiment suggestions outlined by R1, and thoroughly address the points raised by all three reviewers.

We appreciate that these requests represent a great deal of extra work, and we are willing to relax our standard revision time to allow you 6 months to revise your study. Please email us (plosbiology@plos.org) if you have any questions or concerns, or envision needing a (short) extension.

**IMPORTANT - SUBMITTING YOUR REVISION**

*Resubmission Checklist*

*Published Peer Review*

*PLOS Data Policy*

*Blot and Gel Data Policy*

Sincerely,

Taylor

Taylor Hart, PhD,

Associate Editor

PLOS Biology

thart@plos.org

REVIEWS:

Reviewer #1: The manuscript by Ji et al. presents a well-designed study with novel insights into the role of APP in cerebellar motor control. Mechanistically, the authors demonstrated that APP regulates the firing pattern of PCs and inhibitory synaptic transmission from PCs to DCN by modulating Nav1.6 currents, ultimately influencing the motor function. Exogenous expression of APP in Purkinje cells rescues motor-deficits in APP KO mice. In general, the study reveals a novel physiological role for APP and provides new insights into the molecular mechanisms underlying this process. However, I have several concerns and suggestions outlined below.

Major Comments:

1. The present data demonstrated that APP-null mice exhibited reduced locomotor activity and weaker forelimb grip strength compared to WT littermates, and exogenous APP expression in PCs rescued motor deficits in APP-null mice. However, to confirm the essential role of cerebellar APP in motor regulation, it is beeper to perform conditional knockout of APP specifically in PCs.

2. The authors attribute footprint test differences to body size but do not present body weight data. Including body weight comparisons between WT and APP-null mice would strengthen this claim. Moreover, please explain why the rotarod test shows no effects in APP KO or APP KO+APP.

3. In addition to electrophysiological recordings, it is recommended that the authors should examine whether knockout of APP had any effect on the expression level of Nav1.6 in the cerebellum.

4. It is necessary to ask whether Nav1.6 inhibitors/blockers could mimic APP KO, while overexpression of Nav 1.6 (or channel activators/enhancers) could rescue the motor deficits in the APP KO mice.

5. While the study focuses on cerebellar PCs, APP is also expressed in other motor-related regions (e.g., spinal cord). The authors should acknowledge that APP in other neuronal populations might contribute to motor deficits, even if PCs are the primary locus (Page 19, last paragraph).

6. Critical figures lack sufficient resolution or annotations. For example: (1) Include merged image of APP/Calbindin to confirm the colocalization in Figure 1A. (2) Enhance the resolution of Golgi staining images and label dendritic spines/intersections clearly with arrows (or arrowheads) in Supplemental Figure 2. (3) Ensure that all the scale bars, group labels, and statistical annotations (e.g., asterisks) are legible.

Minor Comments:

1. In Page 15, replace "mediated by the Nav1.1-Nav1.6 partnership in PCs" with "mediated by Nav1.1 and Nav1.6 in PCs" for clarity.

2. Define all abbreviations upon first use to improve readability for non-specialists. For instance, “4,9-ahTTX” should be spelled out as “4,9-anhydrotetrodotoxin”.

3. Explicitly state that the study focuses solely on cerebellar PCs and does not explore APP’s role in other motor pathways (e.g., corticospinal tracts). This limitation should be addressed in the Discussion.

4. Fig. 2 and Fig. 5. AP rheobase should also be included into the analysis.

5. Fig. 3D should have a title for the horizontal axis.

6. Fig. 6B. Scale bar: µm, not uM.

Reviewer #2: This study provides an elegant demonstration of the importance of APP in motor control and in cerebellar purkinje cell activity, via modulation of the Nav1.6 sodium channels. The study is overall well performed and experiments are generally well controlled. This brings in vivo confirmation of the interplay between APP and Nav1.6 that the authors highlighted in a previous study in heterologous HEK293 cells. Yet, it would have been a plus if the authors had pushed the study further to bring more information on the molecular mechanism at play. For example, it could have been interesting to identify if full length APP is necessary for the rescue of the cell activity or/and behavioral activity phenotypes or peptides derived from its processing are sufficient, such as for example the C-terminal domain (AICD) or the soluble fragment of APP. Along the same line, the authors do not provide any novel information on how APP regulates Nav1.6 activity beyond what they have published previously in HEK293 cells.

Beyond this first observation, I have some major and minor comments regarding the presented work that should be addressed:

1) The example of APP staining in Figure 1A does not show strong expression of APP in all the cell bodies that are then highlighted with arrows. Could the authors provide better images or a quantification to show APP levels in calbindin positive neurons (using APP null mice as controls)?

2) In their manuscript, the authors explain the decrease in spontaneous firing rate of PC of APP null mice by modifications in sodium Nav 1.6 channel function. However, in Figure 2, together with this decreased firing rate, they observe higher threshold for AP generation and higher AHP. To my opinion, these modifications could also explain the decrease in firing rate. They authors do not address this point. The authors should at least acknowledge this possibility as additional modifications brought about by loss of APP and discuss this point. Please also add 'spontaneous' firing rate because they are not inducing this firing with depolarization steps.

3) Figure 2B : authors describe three profiles of PC activity in WT. Is this expected? If so, they should cite previous work evidencing these different firing patterns.

4) In Figure 3, the authors provide recordings of what they assume are transient, persistent and resurgent sodium currents. To ensure that they are indeed only measuring these currents, they should provide additional data. For example, they should show that these currents are gone in presence of sodium channel blockers.

5) Figure 4 A-D is difficult to understand. In A and C, authors refer to the channel (Nav1.6 or Nav 1.1) in the x-axis, but in C-D authors refer to the drug being used. Also, in C-D, they demonstrate effect in presence of 4,9-ahTTX or ICA by substracting the control currents. What do they mean by this as I don't really understand the nA values reported in C (far right) and D (far right) after this substraction. Would it not be more appropriate to normalize the data in C or D to the average current obtained in ACSF (as 100%) to remove this variable in peak current observed in ACSF WT and APP KO to then correctly assess the drug effect?

6) Last sentence of sub-chapter 4 of Results: the authors write 'These data provide evidence that APP regulates PC firing through Nav1.6'. Where are the data in APP null mice for E-F? How can the authors conclude this from this experiment in E-F?

7) Writing all the results details (numbers, s.e.m., statistics, etc….) within the main text makes the Results section very difficult to read. I would suggest to provide supplementary tables with all these details and only describe the main information for each result obtained with maybe add brief statistics information (e.g. type of test used) in the legend for each Figure panel with statistics.

Minor:

1) Authors should harmonize how they want to call the model: APP null, APP-null or APP KO?

2) Authors should check for punctuation and spaces when providing the reference citations in the text.

3) First paragraph of sub-chapter 2 in Results: authors state 'These results indicated that APP deficiency does not affect the development or degeneration of PCs'. To my opinion, this sentence might be mis-interpreted here. I assume that the authors mean is that, in absence of APP, the morphology of PC cells in normal even in old mice. If so, they should write 'nor provoke degeneration of PCs'.

4) Title of sub-chapter 2 of Results: the title does not make sense - please correct the syntax.

5) Figure 2B: modify red label to read 'silent' not 'slient'.

Reviewer #3: The manuscript by Ji et al. explores the role of Amyloid precursor protein (APP) in motor behavior and cerebellar physiology. The authors demonstrated that motor impairments, such as deficits in grip strength and locomotion observed in APP-null mice, can be rescued by APP gene expression in Purkinje cells (PC). At the cellular level, the authors investigate the impact of APP on PC activity using patch-clamp analysis, with a particular focus on the Nav1.6 and Nav1.1 voltage-gated sodium channels. The manuscript is well-written and offers insights into the involvement of the cerebellum in the pathogenesis of Alzheimer's disease.

Major revisions:

1) In Figure 1 I and J, no significant difference between WT and KO mice is shown. Why did the authors present similar data in Suppl. Figure 1? The only difference between the two sets of figures is the virus injections. Since the main message is already conveyed in Figure 1, the supplementary figure appears redundant and unnecessary.

2) I recommend testing the mice using the beam walking assay to more accurately assess motor coordination and support the conclusion that "the aberrant gait pattern of APP-null mice was due to the body size instead of APP deficiency".

3) Both the results and the figures related to the Nav analysis require improvement. For example, in Figure 4, I recommend displaying the current traces before and after the application of the inhibitors, indicated above the traces. The authors should also include a figure showing the effects of the simultaneous application of Nav1.6 and Nav1.1 inhibitors to better demonstrate the contribution of these channels to the PC Na currents being recorded.

4) Figure 6B needs improvement as the signal appears highly dispersed.

Minor revisions

1) Since the APP-null mice show impairments in the grip test, the statement "these results support the conclusion that APP-null mice exhibit normal neuromuscular functions" is contradictory. Please revise or remove this sentence at the end of the first paragraph of the result section.

2) In Figures 4E and F, clearly indicate on the top of the traces and within the graphs the type of mice being examined.

3) The sentence "we show that the reduction in Nav1.6 sodium currents is a key factor contributing to the motor deficits observed in APP knockout mice" needs revision. This is an assumption rather than a direct cause-and-effect relationship, as the study presents associative findings rather than mechanistic proof.

4) In the methods sections related to laser capture microdissection, specify which fluorescence-positive cells were analyzed.

5) In Figure 2B, change the label to silent.

---

## [Decision Letter · Decision Letter 2]

27 Oct 2025

Dear Dr Liu,

Thank you for your patience while we considered your revised manuscript "Amyloid Precursor Protein Modulates Cerebellar Purkinje Cell Activity and Motor Function via Regulation of Nav1.6 Currents" for publication as a Research Article at PLOS Biology. This revised version of your manuscript has been evaluated by the PLOS Biology editors, the Academic Editor, and several of the original reviewers.

Based on the reviews, we are likely to accept this manuscript for publication, provided you satisfactorily address the remaining points raised by the Academic Editor (see lightly edited comments below the reviews). Please also make sure to address the following data and other policy-related requests.

IMPORTANT: Please ensure that your next revision addresses the following editorial requirements:

-------------------

**Title

We suggest a slight tweak of your title, as we think that the word 'through' is more accessible to non-native speakers than 'via':

"Amyloid Precursor Protein Modulates Cerebellar Purkinje Cell Activity and Motor Function Through Regulation of Nav1.6 Currents"

**Ethics:

-- The Ethics statement needs to be a separate, independent (and the first) subheading in the Material & Methods section. Please separate the following information from the 'Animals' sub-heading: the full name of the IACUC/ethics committee that reviewed and approved the animal care and use, as well as the protocol/permit/project license number. https://journals.plos.org/plosbiology/s/ethical-publishing-practice

**Data:

-- Please cite the location of the data clearly in all relevant main and supplementary Figure legends, e.g. “The data underlying this Figure can be found in S1 Data”.

-------------------

We expect to receive your revised manuscript within two weeks.

*Published Peer Review History*

*Press*

Sincerely,

Taylor

Taylor Hart, PhD,

Associate Editor

thart@plos.org

PLOS Biology

REVIEWS

Reviewer #1: My points raised in the previous round of review have been satisfactorily addressed. Now the paper is acceptable for publication.

Reviewer #3: The revised manuscript shows substantial improvement, and the authors have addressed most of my concerns.

COMMENTS FROM THE ACADEMIC EDITOR (lightly edited):

This new version of the manuscript provides the necessary additional information requested. The new experiments, which were carefully designed, significantly enhance the impact of the manuscript.

I only have one minor comment that would require a sentence in the discussion:

In figure 5B, F and G, the authors show that using a positive allosteric modulator of Nav1.6 the could rescue the firing properties of APP-null PCs and motor deficits in APP-null mice. In the Discussion section, the authors should provide their insights on how this rescue might work mechanistically. In previous work, the authors showed that APP regulates Nav1.6 by enhancing its cell surface distribution. How do the authors link these two observations?'

I also reviewed the responses to the comments of the two other reviewers. It is my personal feeling that the authors made strong efforts to respond to all of them.

---

## [Editor Report · Decision Letter 3]

7 Nov 2025

Dear Dr Liu,

Thank you for the submission of your revised Research Article "Amyloid Precursor Protein Modulates Cerebellar Purkinje Cell Activity and Motor Function through Regulation of Nav1.6 Currents" for publication in PLOS Biology. On behalf of my colleagues and the Academic Editor, Hélène Marie, I am pleased to say that we can in principle accept your manuscript for publication, provided you address any remaining formatting and reporting issues. These will be detailed in an email you should receive within 2-3 business days from our colleagues in the journal operations team; no action is required from you until then. Please note that we will not be able to formally accept your manuscript and schedule it for publication until you have completed any requested changes.

PRESS

Sincerely, 

Taylor

Taylor Hart, PhD,

Associate Editor

PLOS Biology

thart@plos.org